# The RhoA-ROCK1/ROCK2 Pathway Exacerbates Inflammatory Signaling in Immortalized and Primary Microglia

**DOI:** 10.3390/cells12101367

**Published:** 2023-05-11

**Authors:** Elliot J. Glotfelty, Luis B. Tovar-y-Romo, Shih-Chang Hsueh, David Tweedie, Yazhou Li, Brandon K. Harvey, Barry J. Hoffer, Tobias E. Karlsson, Lars Olson, Nigel H. Greig

**Affiliations:** 1Drug Design & Development Section, Translational Gerontology Branch, Intramural Research Program National Institute on Aging, NIH, Baltimore, MD 21224, USA; 2Department of Neuroscience, Karolinska Institutet, 171 77 Stockholm, Sweden; 3Division of Neuroscience, Institute of Cellular Physiology, Universidad Nacional Autónoma de México, Mexico City 04510, Mexico; 4Molecular Mechanisms of Cellular Stress and Inflammation Unit, Integrative Neuroscience Department, National Institute on Drug Abuse, National Institutes of Health, Baltimore, MD 21224, USA; 5Department of Neurosurgery, Case Western Reserve University School of Medicine, Cleveland, OH 44106, USA

**Keywords:** ROCK1, ROCK2, RhoA, microglia, neuroinflammation, ROCK inhibitors, NF-κB

## Abstract

Neuroinflammation is a unifying factor among all acute central nervous system (CNS) injuries and chronic neurodegenerative disorders. Here, we used immortalized microglial (IMG) cells and primary microglia (PMg) to understand the roles of the GTPase Ras homolog gene family member A (RhoA) and its downstream targets Rho-associated coiled-coil-containing protein kinases 1 and 2 (ROCK1 and ROCK2) in neuroinflammation. We used a pan-kinase inhibitor (Y27632) and a ROCK1- and ROCK2-specific inhibitor (RKI1447) to mitigate a lipopolysaccharide (LPS) challenge. In both the IMG cells and PMg, each drug significantly inhibited pro-inflammatory protein production detected in media (TNF-α, IL-6, KC/GRO, and IL-12p70). In the IMG cells, this resulted from the inhibition of NF-κB nuclear translocation and the blocking of neuroinflammatory gene transcription (iNOS, TNF-α, and IL-6). Additionally, we demonstrated the ability of both compounds to block the dephosphorylation and activation of cofilin. In the IMG cells, RhoA activation with Nogo-P4 or narciclasine (Narc) exacerbated the inflammatory response to the LPS challenge. We utilized a siRNA approach to differentiate ROCK1 and ROCK2 activity during the LPS challenges and showed that the blockade of both proteins may mediate the anti-inflammatory effects of Y27632 and RKI1447. Using previously published data, we show that genes in the RhoA/ROCK signaling cascade are highly upregulated in the neurodegenerative microglia (MGnD) from APP/PS-1 transgenic Alzheimer’s disease (AD) mice. In addition to illuminating the specific roles of RhoA/ROCK signaling in neuroinflammation, we demonstrate the utility of using IMG cells as a model for primary microglia in cellular studies.

## 1. Background

Microglia are the resident immune cells of the brain. They perform homeostatic roles in addition to detecting and responding to aberrant local signaling, which leads to their activation. In the past decade, microglia have received increased attention as targets for intervention in brain diseases and injury, particularly the context-dependent phenotypes of these dynamic cells [1]. Some microglia interfere with regenerative or reparative processes by producing large amounts of pro-inflammatory cytokines and chemokines, often referred to as “M1” microglia. Others possess opposite functions, aiding in the repair of damaged or diseased tissue, referred to as “M2” or alternatively activated microglia [2]. This simplified description is useful but is becoming much less relevant, as a spectrum of microglial phenotypes exists regionally in response to injury/age or in the context of neurodegenerative diseases [3]. The distinct microglial gene profiles that define function necessitate a greater understanding of the key internal signaling mechanisms.

Potential new targets for treating neurodegenerative disease and modulating inflammatory signaling in microglia include the small GTPase protein Ras homolog gene family member A (RhoA) and downstream Rho-associated coiled-coil containing protein kinases (ROCK) 1 and 2 [4,5]. RhoA is one of many mammalian Rho GTPases that are known to regulate cytoskeleton dynamics, cell polarity, motility, cell-cycle progression, and vesicle trafficking [6]. RhoA initiates diverse and cell type-dependent signaling [7,8] and does not exclusively activate ROCK1 and ROCK2 [9]. Endogenous signaling from the myelin-inhibiting proteins Nogo, myelin-associated glycoprotein (MAG), and oligodendrocyte myelin glycoprotein (OMgp) and receptors [Nogo-receptor (NgR), sphingosine-1-phosphate receptor 2 (S1PR2), among others] initiate RhoA/ROCK signaling in neurons and other cell types, which is detrimental to neurite repair in injury and disease [10,11,12,13,14]. Some studies have also attributed the pro-inflammatory effects of RhoA/ROCK signaling in microglia to the Nogo family of proteins [15,16].

ROCK1 and ROCK2 are enzymes belonging to the family of serine-threonine kinases, originally discovered for their activity as effector proteins of RhoA [17]. ROCK1 and ROCK2 are 64% homologous, with a 92% homology in their kinase domains [18]. Despite these similarities in kinase regions, there is growing evidence of differing roles for each protein in the periphery and the central nervous system (CNS) [19,20,21,22].

There are a wide variety of pharmacologic tools to inhibit RhoA/ROCK signaling, including over 170 ROCK inhibitors (RIs) [23], most of which are Type 1 adenosine triphosphate (ATP) competitive [24]. RhoA/ROCK signaling in microglia is increasingly seen as a major contributor to the progression of neurodegenerative disorders [5], with RIs reducing microglial activation in animal models of amyotrophic lateral sclerosis (ALS) [25], Parkinson’s disease (PD) [26,27,28], and Alzheimer’s disease (AD) [29]. The RI Y27632-dihydrochloride (Y27632), one of the most ubiquitously studied in this drug class, improves motor performance in animal models of ALS [30], reduces dopaminergic neuron loss among other benefits in mouse models of PD (reviewed in [31]), and facilitates a reduction in toxic amyloid-β (Aβ) protein in a transgenic AD mouse model [32].

Despite the vast number of RIs developed, few have received federal regulatory approval: belumosudil and netarsudil by the Food and Drug Administration (FDA) in the United States and ripasudil and fasudil in Japan, China, and Korea. Belumosudil was approved for the treatment of chronic graft vs. host disease [33], whereas netarsudil and ripasudil are treatments for glaucoma [34]. Fasudil is used for treating vasospasms associated with cerebral ischemia [35] and is one of the earliest RIs approved for clinical use. Several RIs are in clinical trials, including a French trial for ALS (NCT03792490), with others proving that their IV administration is tolerable in humans [36,37]. Together, the wide variety of preclinical animal studies and favorable tolerability of RIs in humans make these drugs candidates for repurposing to treat neurodegenerative disorders.

As microglia are prime targets for drug intervention in the progression of neurodegenerative disease, modeling these cells’ behavior in vitro is essential for drug screening and supplementing in vivo preclinical studies. Immortalized cell lines are useful tools that provide cost-effective models for these purposes. Classic neuroinflammation modeling studies in microglia involve the use of the Gram negative bacterial lipopolysaccharide (LPS), which initiates the activation and nuclear translocation of the transcription factor nuclear factor kappa B (NF-κB) [38]. NF-κB is rapidly mobilized following a stimulus and is a master regulator of inflammatory cytokine gene/protein production (tumor necrosis factor-alpha (TNF-α), interleukin-6 (IL-6), and inducible nitric oxide synthase (iNOS), among others) [39]. In the current study, we use IMG cells [40], a relatively new immortalized microglial cell line, in parallel with primary microglia (PMg) to compare their responses to LPS challenges and treatments with the RIs Y27632- and RKI-1447-dihydrochloride (Y27632 and RKI1447). Y27632 has been widely used in preclinical neurodegenerative studies, while RKI1447 is a newer generation of RI not previously evaluated in microglial-mediated neuroinflammation or animal models of neurodegeneration.

We show here that RhoA activation alone is not sufficient to induce a neuroinflammatory response, but it does exacerbate inflammation when combined with an LPS challenge. By using an siRNA approach, we attempted to discern whether ROCK1, ROCK2, or both proteins are involved in the inflammatory response. Our data suggest that both proteins may affect inflammation. We finally show that neurodegenerative microglia (MGnD) from aged APP/PS1 AD mice (data analyzed from previously published work by Krasemann et al. (2017) [41]) exhibit the upregulation of genes involved in RhoA/ROCK signaling and neuroinflammation. Our study shows that IMG cells may be a viable model for PMg and suggests key roles for RhoA/ROCK signaling in neurodegeneration and neuroinflammation.

## 2. Materials and Methods

### 2.1. Cell Culture Studies

Studies described herein were aimed at assessing the effects of pharmacological interventions targeting ROCK-dependent signaling events in the setting of LPS-induced inflammation. To explore the actions of ROCK activation and inhibition on inflammation, we used two different immunological cell types. We used an immortalized mouse microglial cell line (IMG cells) and freshly isolated primary mouse microglial cultures (PMg). All cell culture studies were performed using standard culturing conditions; namely, the cells were maintained at 37 °C in 5% CO_2_ and 95% air. Specialized cell type-dependent media were used throughout, which were replaced every second day or as indicated.

#### 2.1.1. Immortalized Mouse Microglia (IMG)

The immortalized adherent mouse IMG cells [40] (Sigma-Aldrich; St. Louis, MO, USA; cat# SCC-134, RRID:CVCL_HC49) were cultured in High Glucose DMEM (Sigma Cat. No. D6546) with 10% heat-inactivated fetal bovine serum (Gibco™ cat#10082147), 1× L-Glutamine (Sigma cat# TMS-002-C) and 100 U/mL penicillin/streptomycin (Gibco™ cat#15140148). The subculturing of IMGs was performed as directed in the product datasheet. In brief, cells were treated with Accutase^®^ (Sigma, cat#A6964) for detachment from the plate, collected, and centrifuged, with the cell pellet subsequently resuspended in complete media. The cells were then plated for experiments. IMG cells used in these studies were subjected to no more than 10 passages from the time of thawing.

For drug studies, cells were grown in 24- (40 K cells/well) or 48-well (20 K cells/well) tissue culture-treated plates in 500 μL of media. Fresh media was added on day 1 after passaging and changed every two days as needed. IMG cells used for protein Western blot studies were plated in 6 cm tissue culture-treated dishes at a density of 1 × 10^6^ cells/dish.

#### 2.1.2. Primary Microglia (PMg) Studies

PMg were isolated from young adult (5–7 weeks) male CD-1 mice (Charles River Labs, RRID:IMSR_CRL:022). All mice used in our studies were housed at 25 °C in a 12 h light/12 h dark cycle and given food and water *ad libitum*. All efforts were made to minimize animal suffering and to decrease the number of animals used. All procedures used in this study were fully approved by the Institutional Animal Care and Use Committee (Intramural Research Program, National Institute on Aging, NIH (protocol No. 331-TGB-2024) and followed the NIH guidelines for research using rodents.

PMg cells were prepared as described by Hammond et al. (2019) [42] and Lee and Tansey (2013) [43] with minor modifications. Animals were deeply anesthetized and then transcardially perfused with ice-cold Hank’s balanced salt solution (HBSS, (Gibco cat#14175095). The brains were rapidly removed and placed in HBSS on ice. Individual brains were minced with a scalpel and processed with a 15 mL Dounce homogenizer (Wheaton) using the loose-fitting pestle followed by the tight-fitting pestle (8 strokes with rotation from each pestle). Cell suspensions of approximately 15 mL were filtered on a sterile, pre-wet (HBSS), 100 μm filter (Corning™; Tewksbury, MA, USA; cat# 431751) in 50 mL conical tubes (an additional 5 mL of ice-cold HBSS was used to wash the filter). Filtered suspensions (20 mL) were then centrifuged (600× *g*, 8 min, 4 °C). Supernatant was removed, and cell pellets were resuspended in freshly prepared 37% percoll solution (6 mL) in a 15 mL conical tube. A 2 mL layer of 70% percoll and a 2 mL layer of HBSS was carefully pipetted on the bottom and top of the ~6 mL percoll/cell suspension layer, respectively. Following a 30 min centrifugation (700× *g*, 18 °C, no brake), the interphase layer between 37% and 70% percoll was collected with a sterile fire-polished glass pipette. Cells were washed with HBSS and recentrifuged (300× *g*, 15 min, 4 °C). Cell pellets were then pooled for MACS CD11b staining and separation (Miltenyi Biotec; Gaithersburg, MD, USA; RRID:AB_2654662) according to the manufacturer’s protocol. CD11b positive cells were plated on Poly-D-Lysine (PDL) coated 96-well plates (Corning™; Tweksbury, MA, USA; Biocoat Poly-D-Lysine cellware cat# 354461) at a density of 10,000 cells/well. Cells were allowed to proliferate over 12–14 days in 150 μL complete microglia medium (MM) (ScienCell Research Laboratories; Carlsbad, CA, USA; cat# 1901). Half-volume media changes were made every 2 days for cultures. Ten mice were used for each primary cell culture experiment, with a total of 30 animals used in the current study.

### 2.2. Effects of LPS Challenges and Pharmacological Interventions on ROCK Signaling in Inflammation

#### 2.2.1. Effects of Lipopolysaccharide on IMG and PMg Cells

The effects of a range of concentrations of LPS on the induction of inflammation in IMG cells and PMg were assessed. Cells were challenged with LPS (*E. coli* O55:B5; Sigma, cat# L2880) at the following concentrations: 0.3, 1, 10, 30, 60, 100, and 150 ng/mL. PMg were challenged with LPS preparations (made in MM) at 1, 10, and 100 ng/mL. Cells were cultured in the presence of LPS for 24 h prior to media collection and subsequent downstream analysis of factors secreted into the media.

#### 2.2.2. Pharmacological Interventions Targeting ROCK Signaling in LPS-Induced Inflammation

The effects of Y27632-dihydrochloride (Y27632), a pan-kinase inhibitor, including ROCK1/2 [44] (Tocris cat#1254), and RKI1447-dihydrochloride (RKI-1447), a selective RI (ROCK1/ROCK2) [45], (Tocris cat#5061), were assessed in LPS-activated IMG cells and PMg. RhoA-induced effects on LPS-treated IMG cells were assessed with narciclasine (Narc), an activator of RhoA and downstream ROCK signaling [46] (Tocris cat# 37150), and Nogo-P4, a 25-aa inhibitory peptide sequence and activator of RhoA/ROCK signaling [16], also known as Rat Neurite outgrowth inhibitory peptide (Alpha Diagnostic; San Antonio, TX, USA; cat# Nogo-P4).

In all drug studies, cells were exposed to the compounds/peptide for 1 h prior to a challenge with LPS in the continued presence of drugs/peptide. Submaximal LPS concentrations (10 ng/mL or 2 ng/mL) were used in the studies. In studies with multiple drug/peptide interventions, the drugs were added simultaneously prior to addition of LPS (i.e., Narc + Y277632 or Narc + RKI1447). All drugs/peptides, other than Narc, were prepared in culture media; thus, the appropriate cell culture media was used as a drug vehicle control. As Narc required the use of DMSO to elicit 100% solubility, the drug vehicle for Narc was derived from a DMSO/culture media combination. To confirm reproducibility of drug effects, studies were performed with at least three separate sets of cultures unless otherwise indicated.

#### 2.2.3. Effects of Inhibitors of ROCK Signaling on LPS-Activated IMG Cells and PMg

To assess effects of Y27632 (Tocris cat#1254) in combination with LPS on IMG cells, cells were pre-treated with the drug at 1 μM, 10 μM, 30 μM, 50 μM, 100 μM, 300 μM, and 1000 μM in 500 μL/well. After 1 h, cells were challenged with LPS at 10 ng/mL in the presence of Y27632. Cells were incubated for 24 h, after which conditioned media was then collected and replaced with 500 μL fresh media to allow the assessment of cell viability (described below). Y27632 at doses higher than 100 μM combined with LPS were shown to be cytotoxic (compared to vehicle control); hence, 100 μM of Y27632 was the maximum dose used in subsequent studies. The effects of Y27632 on LPS-activated PMg cells were performed as described for IMG cells. To reduce the numbers of animals used in the studies, cell viability was not assessed separately in PMg. For PMg studies, cells were plated in 96-well plates and incubated in 150 μL media/well ± drug/LPS.

To assess effects of RKI1447 (Tocris, cat# 5061) in combination with LPS in IMG cells, a similar experimental protocol was used as above, with drug concentrations as follows: 500 nM, 1 μM, 10 μM, 30 μM, 50 μM, and 100 μM. After a 1 h pretreatment cells were challenged with LPS (10 ng/mL) in the continued presence of drug. RKI1447 combined with LPS was shown to cause cell toxicity at doses higher than 30 μM, compared to vehicle control. Hence, subsequent experiments use 30 μM as the maximum concentration of RKI1447. For LPS (10 ng/mL)-activated IMG cell plus RKI1447 studies the following drug concentrations were used: 500 nM, 1 μM, 10 μM and 30 μM. As described above, RKI1447 was added 1 h prior to the addition of LPS in the continued presence of drug, for 24 h. PMg were pre-treated with RKI1447 (10 μM and 30 μM) prior to a challenge with LPS (2 ng/mL) and conditioned media was collected 24 h after the addition of LPS.

#### 2.2.4. Effects of Nogo-P4 or Narc on LPS-Activated IMG Cells

IMG cells were plated in 24-well plates and then treated with Narc (Tocris, cat# 3715) or Nogo-P4 at concentrations of 1 nM, 10 nM, 50 nM, 100 nM, 500 nM, or 1 μM (n = 3/group) or 0 μM, 30 μM, 60 μM, or 100 μM, respectively. The highest concentration of Narc (1 μM) contained 0.1% DMSO; accordingly, the vehicle control in this study contained 0.1% DMSO to match this concentration. Doses of Narc above 50 nM were observed to be toxic to the IMG cells (compared to vehicle control). Subsequent Narc studies used a concentration of 50 nM. Narc, once dissolved in DMSO, was not seen to precipitate in cell culture media. No doses of Nogo-P4 were shown to be toxic; hence 60 μM was used in future experiments. Previous studies have used doses of 30 μM [15].

As Nogo-P4 mimics the endogenous RhoA/ROCK signaling of Nogo-66 [47], we first sought to understand if the inflammation response was affected by the peptide, as previously reported [15]. IMG cells plated on 24-well plates were treated with Nogo-P4 (60 μM), LPS (2 ng/mL), or a combination of the two with Nogo-P4 as a 1 h pretreatment. Levels of IMG cell cytokine production, TNF-α, and IL-6, were assessed via ELISA as described below (n = 3/group). Media was replaced with fresh cell culture media, and then cell viability was assessed as described below.

To assess the effects of Narc (50 nM) with inhibitors of ROCK signaling in LPS (2 ng/mL) activated IMG cells, IMG cells were pre-treated with Y27632 (100 μM), RKI1447 (10 μM), and/or Narc. Media only was added to the “Vehicle” and “LPS only” groups for the 1 h pretreatment. Drug pretreatment groups were as follows: Narc only, Narc + Y27632, or Narc + RKI-1447. After 1 h pretreatments, media was removed from the wells and then replaced with vehicle/drugs ± LPS (2 ng/mL). After 24 h, conditioned media was collected for the analysis of secreted factors (TNF-α and IL-6). The media was replaced with fresh cell culture media, and cell viability was then assessed. For MTS study, n = 7 or 8/group. Cytokine assessments were evaluated n = 5 (vehicle) and n = 7 for other treatment groups.

### 2.3. Assessment of Cell Viability with Interventions Targeting ROCK Signaling in LPS-Activated IMG and PMg Cells

Cell viability was quantified using the CellTiter 96^®^ Aqueous One Solution Cell Proliferation Assay kit (MTS) (Promega, Madison, WI, USA, cat#G3580) according to the manufacturer’s protocol. The MTS assay measures the formazan product produced in proportion to the viable cell populations at an absorbance of 490 nm. Following each study, media was collected and replaced with fresh media and assessed for viability. Plates were read using the Tecan infinite M200 Pro plate reader. Results of the MTS assay were used to normalize subsequent cytokine assessments.

### 2.4. Assessment of Media-Secreted Factor Levels following ROCK Signaling Interventions in LPS-Activated IMG and PMg Cells

All secreted factor measurements were normalized to cell viability, assessed via the MTS assay by dividing analyte concentration by optical density (OD) value obtained from the MTS, thus generating a normalized cytokine concentration measurement (normalization example shown in Appendix A). TNF-α and IL-6 concentrations were quantified by use of Enzyme-linked ImmunoSorbent Assays (ELISAs) [BioLegend Mouse TNF-α ELISA MAX™ Deluxe Set (cat#430904); BioLegend, Mouse IL-6 ELISA MAX™ Deluxe Set (cat#431304)]. Assays were performed according to the manufacturer’s protocol. To evaluate the effects of drug interventions on LPS-activated cells we measured TNF-α and IL-6 protein levels in conditioned media. Conditioned media were diluted with the appropriate sample dilution buffer to obtain optical density values in the linear part of the ELISA standard curve. For all studies, vehicle-treated control media were diluted 1:2, and for LPS-activated cells, samples were diluted 1:10 or 1:20. In Narc/Nogo-P4 and siRNA studies, if TNF-α or IL-6 levels were determined to be below the detection limit for each assay, values were defined as 0 pg/mL and are colored red in each graph for clarity. The actions of Narc combined with LPS were confirmed in two separate cell culture experiments. Three separate culture studies confirmed the effects of Y27632 and RKI1447 against an LPS challenge.

A subset of media samples from LPS, Y27632, and RKI1447 treated cells were subjected to multiplex analysis examining the levels of IFN-γ, IL-1β, IL-2, IL-4, IL-5, IL-6, KC/GRO, IL-10, IL-12p70, and TNF-α. The media samples examined were randomly selected from experiments where there were more than three biological replicates in any given treatment group. Media cytokine/chemokine levels were assessed with a V-PLEX Mouse Pro-inflammatory Panel 1, MULTI-SPOT^®^ 96-well, 10-spot plate (cat#N05048A-1, Meso Scale Discovery, 1601 Research Blvd., Rockville, MD 20850-3173, USA), and Meso Quickplex SQ 120 was used to detect the levels of the protein in the media. Media samples where the analyte levels were below the detection limit of the assay were defined as 0 pg/mL and are marked in red on the respective graphs.

### 2.5. Immunochemistry

#### 2.5.1. Immunostaining of Mouse Brain Tissue

Animals were anesthetized with 5% Isoflurane (Forane, Baxter Healthcare Corporation, Deerfield, IL, USA) and perfused transcardially with Phosphate Buffered Saline (PBS) followed by 4% paraformaldehyde in 0.1 M phosphate buffer (PB, pH 7.2). Brains were rapidly removed and post-fixed for 1 day in 4% paraformaldehyde (PFA). Following fixation, brains were sequentially transferred to 20% and 30% sucrose in 1× PBS until the brains sank. The brains were then mounted and frozen onto a cryostat chuck with dry ice using optimal cutting temperature (OCT) compound (VWR cat#361603E). Brains were then cut into 25 μm sections on a cryostat (Leica Biosystems Inc., Buffalo Grove, IL, USA) and placed into cryoprotectant media (30% glycerol, 30% Ethylene glycol, and 40% 1× PBS). For staining, free-floating brain sections were washed with PBS to remove cryoprotectant and then permeabilized/blocked in tissue staining buffer (TSB) (1× PBS with 0.1% Triton-X 100 (SIGMA, Cat # T9284-500ML) and 3% bovine serum albumin (BSA) (Sigma, cat# A7030PBS) in PBS) for 1 h. The brain slices were then incubated in Tris Buffered Saline (TBS) with primary antibodies anti-Iba1 (Synaptic Systems; Göttingen, Germany; cat # 234308, RRID:AB_2924932, dilution 1:500, raised in guinea pig) and anti-TMEM119 (Synaptic Systems, cat # 400 011, RRID:AB_2782984, dilution 1:200, raised in mouse) overnight at 4 °C. Brain slices were then washed three times for 5 min with tissue wash buffer (TWB) (0.1% Triton-X 100 in PBS). Tissue was then incubated for 1 h at RT in BS with the following Alexa-Fluor™ secondary antibodies (Invitrogen; Waltham, MA, USA) (dilution 1:500 in TSB): goat anti-guinea pig-555 IgG (H+L) (cat# A-21435, RRID:AB_2535856) and goat anti-mouse 488 IgG (H+L) (cat# A-11001, RRID:AB_2534069). Tissue was then washed three times for 5 min with TWB and once with PBS. Slices were then mounted on a glass slide using VectaShield with DAPI (Vector Laboratories (Newark, CA, USA), cat# H-1200-10) for nuclei visualization and allowed to cure overnight at 4 °C.

#### 2.5.2. IMG Cells

IMG cells were plated on 12 mm sterile glass coverslips (Fisher Scientific; Waltham, MA, USA; cat#12-545-81) at a density of 30,000 cells/coverslip and grown to ~80% confluency (2–3 days). Coverslips were placed in a non-tissue culture-treated 24-well plate (VWR cat#10861-558) prior to cell addition. Following respective treatments, cells were washed twice with ice-cold PBS and fixed with cold 4% PFA (Fisher Scientific, cat#AAJ19943K2) for 10 min. Following fixation, cells were washed three times with cold PBS and stored in PBS staining at 4 °C. Cells were permeabilized and blocked in microglia staining buffer (MSB) (3% BSA and 0.1% saponin (Sigma, cat#47036) in PBS) for 1 h at room temperature (RT). Cells were incubated in combinations of primary antibodies anti-TMEM119 (Synaptic Systems, cat# 400 011, RRID:AB_2782984, dilution 1:200, raised in mouse), anti-Iba1 (Synaptic Systems, cat#234308, RRID:AB_2924932, 1:500 dilution, raised in guinea pig), and anti-NF-κB (Cell Signaling, cat#6956S, dilution 1:500, raised in mouse) in MSB for 1 h at RT. Following primary antibody incubation, coverslips were washed three times with 0.1% saponin in PBS (wash buffer) for 5 min/wash. Corresponding highly cross-adsorbed Alexa Fluor™ (Invitrogen) secondary antibodies were diluted (1:500) in MSB and applied to cells for 1 h at RT: goat anti-mouse 488 IgG (H+L) (cat# A-11001, RRID:AB_2534069), goat anti-guinea pig IgG 555 (cat#A-21435, RRID: AB_2535856), and goat anti-mouse 555 IgG (cat# A-21422, RRID:AB_2535844). Primary labeled (488 fluorophore) Phalloidin (Invitrogen, cat# A12379, dilution 1:500) staining was performed with secondary antibodies for 1 h at RT. Cells were washed three times in wash buffer for 5 min/wash with a fourth wash with PBS alone. Coverslips were mounted on slides with Prolong Diamond with DAPI (Invitrogen, cat#P36962) (DAPI used to visualize cell nuclei at 405nm wavelength of excitation).

Cells were imaged at 20× or 40× using an LSM 880 confocal microscope (Zeiss, RRID:SCR_020925). A Z-stack was collected through the thickness of each cell culture and collapsed into a maximum intensity projection for the final image (pinhole set to 1 for each channel). For images using NF-κB staining, two to four fields of view (FOV) (1024 × 1024 μm/FOV) were captured/single image using the z-stack and tile scan functions of the Zen software and then used for downstream single cell nuclear NF-κB analysis. Three tile scan images of each coverslip were used for analysis shown (at least 10 FOV per n). A representative single FOV is shown for each experimental group. Imaging parameters were calibrated using secondary-only controls to set laser power and gain.

### 2.6. NF-κB Signal Quantification

Maximum intensity protection TIFF files (with DAPI and NF-κB channels) exported from Zen software were processed and analyzed using FIJI (RRID:SCR_002285) [48] with the following protocol: Separate Channels > Manually threshold DAPI > Fill Holes (DAPI) > Watershed (DAPI) > Analyze Particles (Size: 50-infinity, Circularity: 0.20–1.00, select: Add to Manager) (DAPI) > Select NF-κB Channel Image > Overlay from ROI Manager> Select “Measure” from ROI manager (NF-κB channel image). Mean gray value (MGV) (the sum of the gray values (Optical Density, OD) of all the pixels in each nucleus divided by the total number of pixels/nucleus) was recorded for each nucleus. The MGVs of each nucleus were averaged (~550 cells/n) to generate the average NF-κB signal for each biological replicate. The same procedure was used to measure nuclear NF-κB nucleus signal for the siRNA experiments. Two separate cell culture experiments were performed to confirm the results of these experiments. 

### 2.7. Western Blot Analysis of LPS-Activated IMG Cells

IMG cells were plated on 6 cm dishes and grown to 80% confluency before treatments (Vehicle, Y27632 (100 μM), RKI1447 (10 μM) ± LPS (10 ng/mL)); *n* = 3 or 4 plates/group. Total cell lysates were prepared using ice-cold RIPA buffer (Sigma, cat#RO278-50ML) supplemented with 1× protease/phosphatase inhibitor (ThermoFisher, Halt™ Protease and Phosphatase Inhibitor Cocktail 100×, cat#78842). Cell extracts were agitated for 30 min at 4 °C followed by three one-second rounds of sonication (on ice). Extracts were centrifuged for 20 min (16,000× *g* at 4 °C). Lysates, excluding the DNA pellet, were aliquoted for protein concentration quantification for later use and (stored at −80 °C). Protein concentration for each sample was determined using Pierce™ BCA Protein Acid Kit (ThermoFisher, cat#23225) according to the manufacturer’s protocol. Forty micrograms of total protein were resolved on precast 15-well Nu-PAGE™ (ThermoFisher) 4–12% (cat# NPO323BOX) or 10% (cat# NP0303BOX) Bis-Tris gels and transferred to PVDF membrane, 0.45 μm, (Immobilon, FL-Transfer Membrane, cat # IPFL00010).

Membranes were imaged on an LI-COR imaging system (Odyssey Clx Imager, RRID:SCR_014579, LI-COR Biosciences, 4647 Superior Street, Lincoln, NE, USA) which allows for application of multiplex blotting. Nonspecific binding of proteins to the membrane was prevented by blocking the membrane in Intercept^®^ (PBS) Blocking Buffer (cat# 927-70001) for 1 h at room temperature. The membrane was probed overnight at 4 °C with combinations of the following antibodies diluted in LI-COR blocking buffer: anti-cofilin raised in mouse (Invitrogen, cat# MA5-27737, RRID:AB_2735151, 1:1000 dilution), anti-phospho-cofilin raised in rabbit (Cell Signaling, cat# 331-S, RRID:AB_2080597, 1:1000 dilution) (specifically, cofilin phosphorylation at the Ser3 position), anti-GAPDH raised in mouse (Invitrogen, cat# MA5-15738, RRID:AB_10977387, 1:5000 dilution), anti-GAPDH raised in rabbit (Cell Signaling, cat# 2118S, RRID:AB_561053, 1:1000 dilution), and anti-iNOS raised in rabbit (Cell Signaling, cat# 13120S, RRID:AB_2687529, 1:1000 dilution). The following LI-COR secondary antibodies (diluted 1:20,000) were used for visualization of primary antibody binding: IRDye^®^ 680RD Donkey anti-mouse IgG (cat# 926-68072, RRID:AB_10953628), IRDye^®^ 680RD Donkey anti-rabbit IgG (cat# 926-68073, RRID:AB_10954442), IRDye^®^ Donkey anti-mouse 800CW IgG (cat# 926-32212, RRID:AB_621847), and IRDye^®^ Donkey anti-rabbit 800CW IgG (cat# 926-322-13, RRID:AB_621848). The LI-COR Chameleon™ DUO Pre-Stained Protein Ladder (cat# 928-60000) was used to determine the molecular weight (MW) of proteins. Membranes were imaged using the LI-COR Odyssey imager and analyzed with Image Studio software (Ver 5.2) according to the manufacturer’s recommendations. Results were confirmed in two separate cell culture experiments. Full Western blots from experiments are available in Appendix A.

### 2.8. Assessment of Inflammatory Markers and ROCK Signaling Protein RNA Transcript Levels in LPS-Activated IMG Cells

Total RNA from IMG cells was prepared from single wells of a 24-well tissue plate 24 h post-LPS challenge (n  =  5) using the RNAEasy kit (Qiagen; Hercules, CA, USA; cat# 74136) according to manufacturer’s recommended protocol. Quality and concentration of total RNA were determined using the NanoDrop™ 2000 spectrophotometer (ThermoFisher; Waltham, MA, USA; cat# ND-2000, RRID:SCR_018042) by measuring absorbance at 230, 260, and 280 nm wavelengths. Only RNA within acceptable ratios was used for subsequent analysis. Total RNA (1 μg) was converted to cDNA using the iScript™ cDNA Synthesis kit (20 μL reaction) according to manufacturer’s protocol using a BioRad MJ Mini™ Personal Thermocycler. cDNA generated (10 ng/reaction) was used for downstream RT-qPCR using the iTaq™ Universal SYBR^®^ Green Supermix (Bio-Rad; Hercules, CA, USA; cat#1725121;10 μL reaction).

Primers for iNOS and β-actin used in RT-qPCR experiments were generated using the NCBI primer designing tool. ROCK1 and ROCK2 primers were designed using specified NCBI accession number template in the SnapGene software package (RRID:SCR_015052) (courtesy of Dr. Christopher Richie of the NIH, National Institute on Drug Abuse). The TNF-α and IL-6 primers were previously published as indicated. The sequences and sources of the primer sets used for the RNA transcripts levels are shown in Table 1.

RT-qPCR was run in triplicate (n = 3/group) using the Applied Biosystems (ThermoFisher) QuantStudio 7 Flex Machine and the QuantStudio™ Real-Time PCR Software (RRID:SCR_020245). 40 cycles (PCR stage: 20 s at 95 °C and 20 s at 60 °C; melt curve stage: 15 s at 95 °C, 1 min at 60 °C, and 15 s at 95 °C). Total reaction volume was 10 μL. Transcript levels were expressed as values relative to the control group using the comparative cycle threshold (Ct) method. Relative transcript expression of inflammatory target mRNAs (using the 2^−ΔΔCT^ method) was normalized to β-actin, which was used as an internal control. Results were replicated to confirm reproducibility.

### 2.9. siRNA Experiments

IMG cells were initially plated as described above at a density of 40,000 cells/well (24-well plate). When cells grew to 70% confluency (24 h after plating), IMG media containing 0.5% serum was used to prevent overgrowth during the siRNA transfection and subsequent experiments. Silencer Select pre-designed siRNAs (ThermoFisher) for ROCK1 (cat#4390771), ROCK2 (cat#4390771), and negative control/scrambled moeities (cat#4390843) were used in the experiments. siRNAs were diluted in deionized water upon receipt to the recommended starting concentration of 10 μM, aliquoted, and stored at −20 °C.

Transfection of IMG cells with siRNAs was performed according to manufacturer’s protocol (ThermoFisher protocol publication #MAN0007825 Rev1.0) using Lipofectamine™ RNAiMAX transfection reagent (ThermoFisher, cat#13778075) in OptiMEM™ reduced serum media (Gibco™, cat#31985070) at concentrations for 24-well plates at 5 pmol per siRNA. Since we combined ROCK1 and ROCK2 siRNAs (10 pmol total RNA) for one experimental group, the scrambled siRNA concentration was doubled in the scrambled group (10 pmol total siRNA) and added to wells receiving only ROCK1 or ROCK2 siRNA. This ensured that equal amounts of RNA were used across experimental groups (10 pmol). IMG cells were transfected for 24 h and assessed for knockdown efficiency using RNA extracted as described above and measured using RT-qPCR (also as described above). For siRNA LPS challenge, 2 ng/mL LPS was added to various treatment groups (as described previously) for 24 h. Importantly, media contained siRNAs for the duration of the LPS challenge. Following LPS challenge, the media was collected/stored and replaced with fresh media for MTS assessment. Collected media was later screened for cytokine analysis as previously described and normalized from the MTS assessment.

### 2.10. Statistical Analysis of Data

Data throughout are presented as mean ± standard error of the mean (SEM), with all statistical analysis performed using GraphPad Prism software (version 9.5.0) (RRID:SCR_002798). Statistical outliers were identified by the use of the Grubbs’ test for outliers. Where outliers were identified, they were removed from the analysis and are identified in the text. Data were analyzed with one- or two-way analysis of variance (ANOVA) tests (as indicated) for multiple comparisons and were followed by post hoc tests (Tukey’s or Dunnett’s multiple comparison tests, as indicated). Unpaired t-tests (two-tailed) were employed for two sample comparisons. A value of *p* < 0.05 or less was considered statistically significant. ‘n’ is defined as the number of wells per treatment group.

## 3. Results

### 3.1. Immortalized Microglia (IMG Cells) as a Model for Primary Microglia (PMg)

Iba1 is a traditionally used protein to identify microglia in the brain, although it is also expressed ubiquitously in peripheral macrophages [51]. In recent years, new markers to distinguish between resident brain microglia and blood-derived macrophages have been identified, including transmembrane protein 119 (TMEM119) and purinergic receptor P2Y12 (P2Ry12) [52]. We have previously shown P2Ry12 expression in IMG cells [53], and the current study shows Iba1 and TMEM119 co-expression in IMG cells (Figure 1A). In order to further verify the specificity of the antibodies used to identify the protein expression of Iba1 and TMEM119, we used the antibodies to label a mouse brain, showing high specificity and co-localization (Figure 1A). Secondary-only controls were also used to eliminate any background signal arising from the secondary fluorophores (Appendix A).

In addition to the expression of canonical microglial markers, IMG cells respond to increasing doses of LPS (0–150 ng/mL), with the amplified production of TNF-α, a prominent pro-inflammatory cytokine (Figure 1B(i)). For the IMG cells, TNF-α production peaked with the 100 ng/mL LPS challenge. Similar dose responses to increasing LPS challenges (0–100 ng/mL) were observed in the PMg cultures (Figure 1C(i)). MTS assays for both the IMG cell and PMg LPS dose-response studies are shown in Appendix A. IMG cells exhibit morphological changes when challenged with LPS (Figure 1B(ii)), including increased processes and actin reorganization indicated by phalloidin (filamentous actin) immunoreactivity (Figure 1B(ii)), as was previously described in microglial cultures [54]. PMg also show morphological changes, including increased process length and cell membrane reorganization (Figure 1C(ii)). In vivo mouse models that increase microglial activation, such as TBI [55], show distinct differences between the cultured microglial in our study and those of others’ experiments. Activated, pro-inflammatory microglial typically increase cell body size while decreasing ramification. As cultured cells exist in a two-dimensional environment, they will not perfectly replicate the morphology changes observed in the three-dimensional brain environment.

Further assessments of the vehicle- and 10 ng/mL LPS-challenged IMG cell cultures (Figure 1D) and PMg (Figure 1E) samples confirm a broad spectrum of inflammatory proteins secreted by both. In IMG cells and PMg, TNF-α, IL-6, KC/GRO, IL-10, IL-1β, and IL-12p70 production were significantly increased by the 10 ng/mL LPS challenge. Similar levels of TNF-α, IL-6, and IL-12p70 were produced in both cell types, with PMg generating higher levels of KC/GRO, IL-10, and IL-1β than the IMG cells. Media from the IMG and PMg cells were also probed for IL-4 and IFN-γ, with no measurable analyte present; hence, the data are not shown. As we examined only the submaximal LPS dosage, it may be expected that IMG cells and PMg are capable of producing even higher amounts of each cytokine. The differential production of some cytokines between cell lines and primary microglia has been previously described [56,57].

The drug/toxin models used in our experiments sometimes have marked effects on cell viability, such as the highly mitogenic LPS challenges. For this reason, it is important to normalize the amount of cytokine observed with the viability of each sample, which we have carried out as described in the methods and Appendix A. If significantly more cytokine is present in media, it could be a function of more cells rather than an increased inflammatory effect. For this reason, we normalized all ELISA data to MTS viability (OD) to account for increases or decreases in cell viability.

### 3.2. Pan Kinase Inhibition from Y27632-Dihydrochloride Mitigates Inflammatory Protein Production in IMG and PMg Cells

We next sought to understand the interplay between LPS-induced inflammation and the RhoA/ROCK signaling pathway in both IMG cells and PMg. Figure 2 provides a schematic of LPS-induced NF-κB activation and the interactions with the RhoA/ROCK signaling pathway via toll-like receptor 4 (TLR4) activation. Understanding how ROCK signaling plays into the overall inflammation process is a major goal of current studies. Y27632 is a pan-kinase inhibitor, enacting the inhibition of kinases via competitive binding (against ATP) to the catalytic region of the enzymes [58]. Y27632 inhibits ROCK1, ROCK2, protein kinase A (PKA), protein kinase C (PKC), and myosin light chain kinase (MLCK) but is typically referred to as an RI due to its higher specificity to the two ROCK enzymes (Figure 3A). A cell viability assay of Y27632 with LPS was first performed to determine the optimal dosing for our assays. We observed significant toxicity from Y27632 (vs. vehicle treatment) in the two highest doses assessed: 300 μM and 1000 μM (Appendix A); hence, the maximal dosing of 100 μM was used throughout our studies. In both the IMG cells and PMg, a 1 h pretreatment of Y27632 reduced the LPS (10 ng/mL)-induced production of TNF-α in a dose-dependent manner. Significant reductions in TNF-α were observed at doses as low as 10 μM for the IMG cells and 50 μM for the PMg (Figure 3B). IMG cell and PMg TNF-α production was normalized to cell viability (determined via MTS, shown in Appendix A). Samples from Figure 3B (100 μM Y27632 for IMG cells; 50 μM Y27632 for PMg and LPS only) were analyzed using a multiplex cytokine analysis (Figure 3C,D). In the IMG cells, the multiplex analysis confirms significant reductions in TNF-α (68.5%; *p* < 0.0001) as well as the pro-inflammatory cytokine/chemokines IL-6 (91%, *p* < 0.0001), KC/GRO (86%; *p* < 0.0001), and IL-12p70 (77.4%; *p* < 0.0001) when compared to LPS challenge alone (Figure 3C). The Y27632-treated IMG cells show a minor, albeit significant, increase in the production of IL-1β (*p* = 0.03930) when compared to LPS challenge alone. Similar trends in Y27632 (50 μM)-treated, LPS-challenged PMg (n = 3/group) were observed in the multiplex analysis. Significant TNF-α reductions (48%; *p* = 0.0293) were confirmed, with decreases in IL-6 (72.5%; *p* = 0.0026) and IL-12p70 (70%, *p* < 0.01) also observed (Figure 3D). Y27632 treatment also decreased KC/GRO production in PMg (28.2%), although the result was not significant (*p* = 0.3184). PMg also showed significant reductions in the lower expressed proteins IL-2 (*p* = 0.0294) and IL-1β (*p* = 0.0058). The anti-inflammatory cytokine IL-10 significantly increased (*p* = 0.0015) in 50 μM Y27632-treated PMg. Although the multiplex assay used in C and D also measured IL-4 and IFN-γ, no measurable analyte was observed and, hence, is not shown.

We further analyzed the whole-cell IMG lysates via Western blotting to investigate the intracellular inflammatory protein content, specifically iNOS, a well-described inflammatory protein induced by toll-like receptor 4 (TLR4) activation via LPS. The overproduction of iNOS leads to an intracellular buildup of nitric oxide, a free radical that can become cytotoxic [59]. Our experiments show the LPS (10 ng/mL) challenge increases iNOS production in IMG cells, which is significantly reduced (*p* = 0.0037) by 100 μM Y27632 treatment, restoring the cells to the vehicle levels of the protein (Figure 3E). This effect was observed in three separate cell cultures, ensuring reproducibility.

### 3.3. ROCK1 and ROCK2 Inhibition Is Sufficient to Blunt Inflammatory Protein Production in IMG Cells and PMg

In order to investigate whether ROCK1 and ROCK2 manipulations are sufficient to reproduce the anti-inflammatory properties of Y27632, we used a highly specific ROCK1 and ROCK2 inhibitor, RKI447 dihydrochloride (RKI1447) (Figure 4A), against LPS-challenged IMG cells and PMg. RKI1447 does not inhibit the wide range of kinases that Y27632 does, and thus, the activity can be more directly linked to the inhibition of ROCK1 and ROCK2 [45]. We first determined the toxicity of RKI1447 with LPS (10 ng/mL) in IMG cells to define the optimal dosing of the drug. At doses above 30 μM (the maximum dose used in our studies), we found that RKI1447 is highly toxic to IMG cells, resulting in nearly 50% cell death (50 μM and 100 μM RKI1447) (MTS data are shown in Appendix A). Similar to our previous experiments with Y27632, RKI1447 decreases TNF-α production following an LPS challenge (10 ng/mL) in a dose-dependent manner in IMG cells. At 10 μM (28%; *p* = 0.0466) and 30 μM doses (78%; *p* < 0.0001), significant decreases were observed (Figure 4B) (MTS is shown in Appendix A). In PMg, a 30 μM RKI447 treatment against 10 ng/mL LPS, likewise, significantly decreased TNF-α production (40.4%; *p* = 0.0011; (Figure 4B) (MTS data are shown in Appendix A). The multiplex analyte analysis of 30 μM pre-treated IMG cells under LPS challenge (from Figure 4B) (n = 3) confirmed significant reductions in TNF-α (71%; *p* < 0.0001) as well as the pro-inflammatory cytokine/chemokines IL-6 (99%; *p* < 0.0001), KC/GRO (60.1%; *p* = 0.004), and IL-12p70 (100%; *p* = 0.0012). Minimally secreted IL-1β (*p* = 0.0006) and the anti-inflammatory IL-10 (*p* = 0.0023; Figure 4C) also significantly decreased. Similar trends for the 30 μM RKI1447-treated PMg + LPS cells (Figure 4D) (n = 3) were observed via the multiplex analyte analysis, and we validated significant TNF-α reductions (41.6%; *p* = 0.0016) together with significant decreases in IL-6 (86.56%; *p* < 0.0001), KC/GRO (63%; *p* = 0.0004), IL-12p70 (100%; *p* = 0.002), IL-1β (98.56%; *p* = 0.0001) and the anti-inflammatory IL-10 (76%; *p* = 0.0005). Although the multiplex assay used in C and D also measured IL-4 and IFN-γ, no measurable analyte was observed from any treatment, and, hence, neither is shown.

The whole-cell IMG cell lysates probed via Western blotting for iNOS, normalized to GAPDH protein, closely match the results observed in the Y27632 treatments (Figure 4E). Significant reductions in iNOS protein were observed when compared to 10 ng/mL LPS treatment (72.12%; *p* < 0.0001). Thus, overall, RKI1447 is similar to Y27632 in its ability to reduce cytokine and related inflammatory protein production.

### 3.4. Y27632 and RKI1447 Restore Ratios of Inactive Cofilin to Total Cofilin in IMG Cells Challenged with LPS

Cofilin plays a dynamic role in actin filament binding, severing, and depolymerizing activities and is downstream of the ROCK signaling pathway. The activation of ROCK induces the phosphorylation of cofilin (at the Ser3 position); however, cofilin is only active in its dephosphorylated state [60]. Recently, cofilin was shown to mediate the LPS-induced inflammatory response in a spontaneously immortalized microglial cell line (SIM-A9) [61]. Additionally, cofilin knockdown or inhibition in a mouse model of hemorrhagic brain injury improved behavioral outcomes, reduced oxidative stress, and decreased microglial activation [62]. The Western blot analysis of LPS-treated IMG cells without Y27632 (*p* = 0.0004) or RKI1447 (*p* < 0.0001) treatment shows significant decreases in the ratio of phosphorylated cofilin to total cofilin (~40%) (Figure 5A–C). Both Y27632 (100 μM) (*p* = 0.0011) (Figure 5B) and RKI1447 (30 μM) (Figure 5C) restore the ratio of phosphorylated cofilin/total cofilin to vehicle treatment levels. As ROCK activation results in the increased phosphorylation of cofilin, ROCK inhibition likely blocks an alternative pathway that has not been currently explored (Figure 5D).

### 3.5. RhoA Activation Is Not Sufficient to Initiate an Inflammatory Response in IMG Cells but Exacerbates Inflammation in the Presence of a Submaximal LPS Challenge

In order to further understand and verify the influence of ROCK signaling on the inflammation process, we used Narc and Nogo inhibitory peptide (Nogo-P4) pretreatments, with and without LPS (2 ng/mL) challenges, in IMG cells. Narciclasine (Narc) is a plant growth inhibitor that potently activates RhoA, which is necessary for ROCK1/2 activation [46] (Figure 6A). Nogo-P4 is a commercially available peptide that contains the Nogo-receptor-1 (NgR1) activating sequences of the endogenous Nogo-A, -B, and -C (Nogo-66). Nogo-P4 acts similarly to Nogo-66, which is known to activate RhoA (Figure 6A) in neurons [63] and in microglia [16].

We first assessed the toxicity of various doses of both Nogo-P4 and Narc using the MTS assay. We did not observe any toxicity associated with the doses of Nogo-P4 up to 100 μM (Figure 6B(i)), although Narc was shown to be potently toxic at doses as low as 100 nM (Figure 6B(ii); *p* < 0.0001). Contrary to a prior study indicating the pro-inflammatory action of Nogo-P4 alone in microglial cells [15] at a dosage of 32 μM, we did not find increases in TNF-α (Figure 6D(i)) or IL-6 (Figure 6D(ii)) concentrations in media associated with an even higher dose of Nogo-P4 (60 μM) alone when compared to the vehicle treatment. However, combined with an LPS challenge (2 ng/mL), Nogo-P4 significantly increased the amount of TNF-α produced in IMG cells (Figure 6D(i)) compared to LPS alone (*p* = 0.0035). We next assessed effects of the LPS (2 ng/mL) ± pretreatments of Narc (50 nM) ± RKI1447 (10 μM) or Y27632 (100 μM) in IMG cells. The MTS data are shown in Appendix A. Narc (50 nM) alone caused a 12.1% decrease in cell viability, as assessed via MTS, when compared to the vehicle-treated cells (Figure 6E). When combined with an LPS challenge (2 ng/mL) ± RKI1447 (10 μM) or Y27632 (100 μM), cell viability was restored to vehicle levels. The IMG cells treated only with LPS showed a nearly 30% increase in viability compared to the vehicle-treated cells (Figure 6E). Grubbs’ test for outliers identified one outlier in the vehicle group and one in the “LPS only” group. These samples were removed from the analysis and were not further analyzed for cytokine content. We further analyzed the effects of these treatments on inflammatory protein production (TNF-α and IL-6) and showed significantly exacerbated TNF-α (~4-fold increase; *p* < 0.0001, Figure 6F(i)) and IL-6 (~1.23-fold increase; *p* < 0.0001, Figure 6F(ii)) production from the IMG cells resulting from the LPS + Narc treatment when compared to the LPS treatment alone. When Narc (50 nM) was combined with RKI1447 (10 μM) or Y27632 (100 μM) as a pretreatment with LPS challenge, exacerbated inflammatory protein production (TNF-α and IL-6) was significantly reduced or abolished (*p* < 0.0001, compared to LPS + Narc). As with the Nogo-P4, Narc alone (50 nM) did not induce increases in TNF-α (Figure 6F(i)) or IL-6 (Figure 6F(ii)). Our study is in line with others showing that RhoA specifically plays a role in neuroinflammatory processes [64,65].

Microglial morphology provides hints for the activation states of these cells. Homeostatic microglia (as shown in Figure 1A) are ramified and perform maintenance roles in the brain. When homeostatic microglia encounter damaged tissue or are exposed to injury or disease, they become activated and shift morphology to a round or amoeboid state [55,66]. These cells are highly phagocytic and are capable of producing pro-inflammatory cytokines [67]. Representative images of the IMG cells from each treatment group, assessed in Figure 6E,F, are shown in Figure 6G. Although the IMG cells are not as ramified as resident microglia in the brain (Figure 1A), the untreated IMG cells retain the processes that are lost upon Narc and Narc + LPS treatment, which is indicative of an activated, strongly pro-inflammatory phenotype. The LPS-only-treated IMG cells are highly proliferative with a wide variety of morphologies, including cells with processes. Y27632 (100 μM) and RKI1447 (10 μM) robustly restore the IMG cell processes lost by the Narc + LPS challenge, reflecting a phenotypic shift more closely resembling the vehicle-treated cells.

### 3.6. Y27632 and RKI1447 Block the Nuclear Translocation of NF-κB in IMG Cells

In order to further understand the mechanism of action of Y27632 and RKI1447, we used IMG cells to assess the levels of NF-κB, a key protein in both innate and adaptive immunity, that translocates to the nucleus of stimulated cells where it performs its biological function. NF-κB plays an essential role in the initiation of inflammatory gene transcription and downstream protein production [68]. Maximal NF-κB nuclear translocation occurs rapidly from TLR4 activation via LPS in less than 20 min [69] (Figure 2). NF-κB oscillates over a larger time scale in vitro, which is a mechanism to monitor environmental changes from the initial translocation event [70,71].

Here, we first treated the IMG cells with either Y27632 (100 μM) or RKI1447 (10 μM) for 1 h prior to the LPS challenge (10 ng/mL), as in our previous experiments. A schematic of the experimental setup is pictured in Figure 7A. The confocal images (Figure 7B) were exported and analyzed in FIJI for single-cell nuclear NF-κB signals (n = 3/group). A 6.4-fold increase (*p* < 0.0001) in the NF-κB signal resulted from the LPS treatment alone, with both Y27632 (100 μM) and RKI1447 (10 μM) blocking this effect (Figure 7C). The individual nuclei collated from each group (from Figure 7C) were plotted on a histogram and showed a clear shift in the signal-to-vehicle treatment levels (Figure 7D). We next used RT-qPCR to measure the mRNA levels of three genes that NF-κB regulates (TNF-α, IL-6, and iNOS) at 24 h post-LPS incubation (Figure 7E). Y27632 (100 μM) and RKI1447 (10 μM) significantly attenuated LPS-induced increases in IL-6 (*p* < 0.0001) and iNOS (*p* = 0.0107 and *p* = 0.0367) mRNA. Both Y27632 and RKI1447 reduced TNF-α mRNA levels (*p* = 0.0018 and *p* = 0.0738, respectively).

### 3.7. Evidence for ROCK1 and ROCK2 Roles in LPS-Induced Inflammatory Response

Although evidence exists for the differential roles of ROCK1 and ROCK2 [19,20,21], the separate contributions of these proteins to the inflammation process have not been previously explored. We used a siRNA approach using IMG cells to assess the roles of ROCK1 and ROCK2 in an LPS-induced inflammatory challenge. Both ROCK1 and ROCK2 are expressed similarly in IMG cells (Figure 8A). ROCK1 and ROCK2 siRNAs applied for 24 h are specific and significantly knock down the target genes of ROCK1 (37.2% decrease, *p* = 0.0014) and ROCK2 (36.6% decrease, *p* = 0.0016) when combined in the same treatment (40.6% decrease, *p* = 0.0006 and 29% decrease, *p* = 0.0109, respectively) vs. scrambled siRNA control-treated cells’ expression levels (Figure 8B). Following 24 h of siRNA treatment, the IMG cells were challenged with 2 ng/mL LPS for 15 min and probed for NF-κB protein localization, as in Figure 7. Representative images of these cells are shown in Figure 8C. The non-LPS-challenged scrambled siRNA-treated IMG cells have a significantly lower (*p* = 0.0007) nuclear NF-κB signal than the LPS-challenged cells. Notably, this baseline signal is higher than the non-siRNA-treated vehicle control cells in Figure 7. In the IMG cells challenged with LPS (2 ng/mL), the average nuclear NF-κB signal was significantly reduced by ROCK2 (*p* = 0.0144) and ROCK1 + ROCK2 (*p* = 0.0102) siRNA treatments when compared to scrambled siRNA. ROCK 1 siRNA treatment decreased the average nuclear NF-κB signal, almost reaching significance (*p* = 0.0897; Figure 8D). All individual IMG cells’ average nuclear NF-κB signals (from Figure 8D) are shown via a histogram, with lower signal distribution for the ROCK1, ROCK2, and ROCK1 + ROCK2 siRNA + LPS treatments when compared to scrambled siRNA + LPS treatment (Figure 8E). In addition to assessing NF-κB protein localization in the IMG cells, we also analyzed the effects of siRNA treatments on IL-6 (Figure 8F) and TNF-α (Figure 8G) production 24 h post-LPS challenge. IL-6 production from the LPS-challenged cells decreased only with ROCK1 + ROCK2 siRNA treatment (33.2% reduction, *p* = 0.0075) when compared to the scrambled siRNA treatment. TNF-α production was not affected by the siRNA treatments. The MTS assay is shown in Appendix A.

### 3.8. Neurodegenerative Clec7a^+^ Microglia Express Increased RhoA, Cofilin, and the Key Modulators of Neuroinflammation

As shown herein, microglia are a dynamic cell type that can alter the phenotype and gene expression profiles in a context-dependent manner. Clec7a is a gene that was previously reported as a marker of disease-associated microglia (DAM [72]) lipid droplet accumulating microglia (LDAM [73]); and neurodegenerative microglia [41,74], with the latter two reported to be very detrimental to disease progression in mouse models of AD. Interestingly, all of these types of microglia possess distinct genetic profiles [73]. Previous studies have demonstrated that microglia respond to amyloid-β (Aβ) challenge by expressing a pro-inflammatory phenotype [75]. Here we show the previously published RNAseq (SMART Seq2 profile) mRNA expression levels of key RhoA/ROCK signaling genes [RHOA, ROCK1, ROCK2, and CFL1 (Cofilin)] and the regulators of inflammatory responses (CCL3 and CCL4) from microglia isolated from AD transgenic mice (both Clec7^+^ (MGnD) and Clec7^−^, and similarly aged (9 months) WT C57BL/J6 mice (Figure 9A; data from [41]). The AD mice, known as APP/PS1 mice, are double transgenic for mutations associated with early-onset AD, expressing a chimeric mouse/human amyloid precursor protein (APP) (Mo/HuAPP695swe) and a mutant human presenilin 1 (PS1-de9). Clec7a^+^ microglia are associated with amyloid-β (Aβ) plaques (Figure 9A), whereas the Clec7a^−^ microglia are localized outside of plaque formations. The Clec7a^−^ microglia from the APP/PS1 mice show no difference in the expression of RhoA, ROCK1, ROCK2, CFN1, CCL3, or CCL4 (Figure 9B–G) and show overall similar expression profiles to the microglia from WT, aged mice (from Figure 2C of [41]). RhoA, CFN1, CCL3, and CCL4 expression are significantly upregulated in Clec7a^+^ APP/PS1 “neurodegenerative” microglia when compared to WT microglia (*p* = 0.0020, *p* < 0.0001, *p* < 0.0001, and *p* < 0.0001, respectively) and Clec7a^−^ APP/PS1 microglia (*p* = 0.0018, *p* < 0.0001, *p* < 0.0001, and *p* < 0.0001, respectively; Figure 9B,E–G). ROCK1 and ROCK2 are minimally expressed, with ROCK1 showing no changes among the three subtypes of microglia, with ROCK2 showing small but significant downregulation relative to other transcripts (mean expression in WT: 9.305 FPKM, APP/PS1 Clec7a^−^: 9.72 FPKM, and APP/PS1 Clec7a^+^: 6.261 FPKM) compared to WT (*p* = 0.0221) and APP/PS1 Clec7a^−^ (*p* = 0.0098) microglia (Figure 9C,D).

## 4. Discussion

As microglia are the source of both the maladaptive and beneficial aspects of neuroinflammation, they are an important target for disease-modifying CNS drugs, of which there are few that are effective. Importantly, our current work expands the understanding and potential utility of the IMG cell line originally characterized in 2016 [40], adding to the existing tools and cell models used in anti-inflammatory CNS drug development. The BV-2 microglial cell line is the most used cell line for modeling microglia [76]. Although BV-2 cells have proven useful for modeling microglia, there are some differences when compared to PMg, such as higher baseline, non-activated levels of secreted cytokines, reactivity to LPS, and morphological differences [77]. New cell lines, such as IMG cells, may be better suited to model PMg, as they have been shown to express canonical microglial markers (*Fcrls*, *Sall1,* and *Itgb5*) at significantly higher levels than in BV-2 cells [40]. Microglial cell cultures, both primary and immortalized, behave differently from those in the brain [78], and these limitations should be considered within in vivo microglial experimentation. As noted by Bohlen et al. (2019) [79], microglial cultures cannot be expected to predict in vivo microglial behavior, though they remain useful tools for isolated cell behavior and dynamics.

When compared to PMg, which require the use of many mice or rats, accompanied by a time-consuming extraction procedure, IMG cells are relatively inexpensive and provide the capacity to perform many experiments from a small number of starting cells. We show that IMG cells express the canonical markers of resident brain microglia, including TMEM119, along with the traditionally used Iba1. We have previously shown that IMG cells also express the purinergic receptor P2Ry12 [53]. TMEM119 and P2Ry12 are important microglial markers, as these proteins are not expressed in peripheral macrophages. IMG cells and PMg respond to LPS challenges with changes in morphology and dose-dependent increases in cytokine production. Similar cytokine production profiles in response to LPS challenges were observed between the two cell types.

Our observation that pan-kinase (inclusive of ROCK1 and ROCK2) inhibition via Y27632 pretreatment effectively mitigates LPS-induced inflammation in IMG cells and PMg is in line with studies showing the anti-inflammatory effects of Y27632 [28,80,81,82], but this provides more detailed information regarding the overall cytokine production effects of the drug against an LPS challenge. We further show that ROCK1- and ROCK2-specific inhibition via RKI-1447 [45] is sufficient to mitigate pro-inflammatory cytokine production in both IMG cells and PMg.

In order to understand the mechanisms of the anti-inflammatory effects of both RIs, we used IMG cells. We show, for the first time in microglial cells, that Y27632 and RKI1447 mediate the blockade of the nuclear translocation and activation of NF-κB induced by LPS as a primary anti-inflammatory mechanism. RIs, including Y27632 but not RKI1447, have previously been shown to block the activation and nuclear translocation of NF-κB in a wide variety of other cell types [65,83,84,85]. As NF-κB is a master regulator of inflammatory cytokine gene transcription [39], we confirmed the decreases in TNF-α, IL-6, and iNOS mRNA from the Y27632 and RKI1447 treatments against an LPS challenge.

Cofilin is one of the terminal effector proteins of RhoA/ROCK/LIMK signaling, resulting in the phosphorylation and inactivation of the protein. Active cofilin is implicated in disease/injury pathology in PD, AD, and stroke (reviewed in [86]), with microglial cofilin playing a role in age-related neuroinflammation [87]. The phosphorylation of cofilin at the Ser3 position renders the protein inactive, thus reducing actin turnover, whereas phosphatases, such as slingshot (SSH), restore the activation of the protein [88]. Cofilin knockdown using siRNA resulted in decreased cytokine release, including TNF-α and nitrite, as well as inflammation-related protein expression (iNOS and COX2) in LPS-stimulated microglia [61]. Recently, a first-in-drug-class cofilin inactivator was developed that had similar anti-inflammatory effects in an in vitro model of stroke [89].

Blocking ROCK1 and ROCK2 would infer that less cofilin will be phosphorylated, which is true on shorter timescales (<6 h) [90]; however, we show here that ROCK inhibition with Y27632 and RKI1447 against an LPS challenge restores the basal levels of phospho-cofilin after 24 h when compared to the LPS treatment alone. This indicates that ROCK signaling is also interacting with other signaling proteins that affect cofilin phosphorylation during the LPS challenge. Nucleotide-binding oligomerization domain-containing protein 1 (NOD1), which contributes to the immune response, might be a likely candidate. NOD1 contributes to the LPS-mediated inflammatory response [91], is intimately linked to SSH activity, and is affected by ROCK signaling [92]. Further mechanistic studies are needed to fully understand how RIs confer their anti-inflammatory properties, but it is clear that the RIs used in the current study effectively reverse LPS-induced changes in cofilin phosphorylation.

In this study, we observed that IMG cells treated only with the RhoA activators Nogo-P4 and Narc did not respond with inflammatory protein production. This is contradictory to a previous report on the pro-inflammatory effect of Nogo-P4 alone in BV-2 microglia [15] but is in line with a previous study that showed that Narc alone did not induce an inflammatory response [93]. The differences in how Nogo-P4 was applied (coated on wells rather than added in solution) in Fang et al. (2015) [15] may account for the differences in the observed response.

Both Nogo-P4 and Narc pre-incubations exacerbated LPS-induced cytokine production. In the Narc + LPS-treated IMG cells, both RKI1447 and Y27632 significantly decreased the exacerbated inflammatory effects of Narc + LPS, implying that increases in cytokine production are RhoA/ROCK-mediated. Our results from the Narc treatments are contradictory to other studies in which an anti-inflammatory effect from Narc was observed [46,94,95,96]. The dosages of Narc used in the Zhao et al. (2021) [94] studies in BV-2 microglia against LPS challenge ranged from 2- to 6-fold higher to those used in our studies. In addition, the normalization of cell viability was not applied throughout, and thus the two studies are not directly equivalent. It is also possible that BV-2 cells react differently than IMG cells, but further studies are needed to assess the comparability of these cell types.

Our results are in line with a previous work in human neutrophils, showing that RhoA has a dual role depending on the activation state of the cell. In a resting cell, RhoA inhibits inflammation, but in an LPS-challenged cell, RhoA promotes inflammation via ROCK [97]. Because RhoA signals through multiple downstream pathways [98] and might be context-dependent, as described in [97], the potential roles of RhoA and related proteins in inflammation should be further evaluated in microglia. Interestingly, the genetic ablation of microglial-specific RhoA in mice increased the microglial production of inflammatory proteins and elicited an AD-like pathology [99]. The differences observed in the literature regarding RhoA’s role in inflammation may be from the many downstream pathways the protein influences.

Endogenous RhoA/ROCK activators with relevance to the current studies, including the reticulon proteins (Nogo-A, B, and C), are modeled here with Nogo-P4. Nogo-P4 mimics the active regions of all three Nogo isoforms, specifically the extracellular loop consisting of 66 amino acids (Nogo-66). In the human brain, the increased expression of Nogo protein has been observed in patients with AD [100], multiple sclerosis [101], and temporal lobe epilepsy [102]. Although Nogo is implicated in these human conditions, decreases in Nogo-A mRNA have been observed in aged rats [103] and mice [104]. Our studies demonstrate that increased RhoA activation may exacerbate inflammation. Notably, the use of anti-Nogo antibody therapies has been effective in improving recovery in animal models of stroke [105] and spinal cord injuries [106]. Cerebrospinal fluid (CSF) concentrations of RTN4 (Nogo) were found to be significantly elevated in patients with AD, PD, and MS compared to controls, highlighting a connection between neurodegeneration and an endogenous RhoA activator [107].

Despite the similarities in the kinase regions of ROCK1 and ROCK2, there is growing evidence of the differing roles of each protein, including in the brain [19,20,21,22]. We sought to address whether ROCK1 or ROCK2 inhibition had a more dominant role in the anti-inflammatory properties of Y27632 and RKI1447. ROCK2 is more enriched in the brain than ROCK1 [108]; however, Zhang et al. (2016) [109] (www.brainrnaseq.org, accessed 1 December 2022) showed that human microglia are a ROCK1-enriched niche in brain cells, expressing over double the amount of ROCK2 mRNA. We show here that ROCK1 and ROCK2 are similarly expressed in IMG cells and used an siRNA approach to downregulate both genes. ROCK1, ROCK2, and ROCK1+ROCK2 siRNA treatments reduced LPS-induced NF-κB nuclear translocation when compared to the control group, with ROCK2 and ROCK1+ROCK2 significantly doing so. This implies both ROCK1 and ROCK2 have a role in LPS-mediated inflammation response. IL-6 production from an LPS challenge was significantly reduced only from the ROCK1+ROCK2 siRNA treatment, again implying a synergistic role of ROCK1 and ROCK2 in the inflammation response.

Considering the wide range of context-dependent phenotypes microglia possess [42], we probed previously published RNAseq data from nine-month-old mice that contained a neurodegenerative subtype of microglia (MGnD) associated with Aβ plaques in a mouse APP/PS1 AD model [41]. This subtype of microglia exacerbates neurodegeneration and is differentiated from other types of microglia due to the expression of the protein Clec7a and its localization to Aβ plaques. Here, we show that MGnD express significantly higher levels of RhoA, CFL1 (cofilin), and the inflammation-regulating genes CCL3 and CCL4 when compared to Clec7^−^ microglia from the same mice and age-matched WT controls. ROCK1 expression remained stable throughout, with ROCK2 interestingly downregulated in MGnD. These findings further support the need to treat microglia as a heterogeneous cell population and suggest that RhoA/ROCK signaling may also play a role in the pathological phenotypes of these cells.

Our studies show that RIs may be a suitable drug class to repurpose for use in neurodegenerative disorders. Of paramount concern is a drug’s ability to penetrate the blood-brain barrier (BBB), and this is one of the greatest challenges to overcome in treating brain disorders or injuries [110]. With the wide variety of RIs developed, this consideration is particularly important in terms of assessment in preclinical studies. Little work has been undertaken to compare BBB penetration between RIs [111], but there is evidence that fasudil can enter the brain, albeit at low concentrations [25,112]. The direct brain infusion of fasudil has also been used in a mouse model of stroke with positive outcomes [113]. Even with low levels of brain penetration, fasudil was able to reverse the cognitive deficits associated with an AD mouse model [112]. RIs have been proven safe in humans, but their application to diseases of the brain remains understudied.

Despite the large body of research encompassing RIs, there remains much to be studied regarding this signaling pathway. Exploring these drugs’ effects in vivo will be an important next step. Overall, our work indicates that ROCK proteins are viable targets for intervention in brain diseases and injuries due to their potent effects on the inflammation process.

## 5. Conclusions

Herein, we illustrate the importance of the RhoA/ROCK signaling pathway in microglia-derived inflammation. Neuroinflammation plays a role in nearly all neurodegenerative diseases and injuries, and the dearth of drugs available for treating these conditions increases the urgency for new compounds to be investigated. Here, we demonstrate the potent anti-inflammatory capacity of Y27632 and RKI1447 in both IMG cells and PMg and validate the utility of using IMG cells as a model for PMg. The use of new cell lines will become increasingly important as the FDA has dropped the requirements for animal models to be used in drug development [114]. IMG cells may fill an important place in drug development that targets neuroinflammation. While much focus has been put into studying the roles of ROCK2 in the brain, our studies show that more focus should, likewise, be put into studying the roles of ROCK1, especially in neuroinflammation. Lastly, subtypes of microglia may confer pro-inflammatory phenotypes via the RhoA/ROCK signaling system.

## Figures and Tables

**Figure 1 cells-12-01367-f001:**
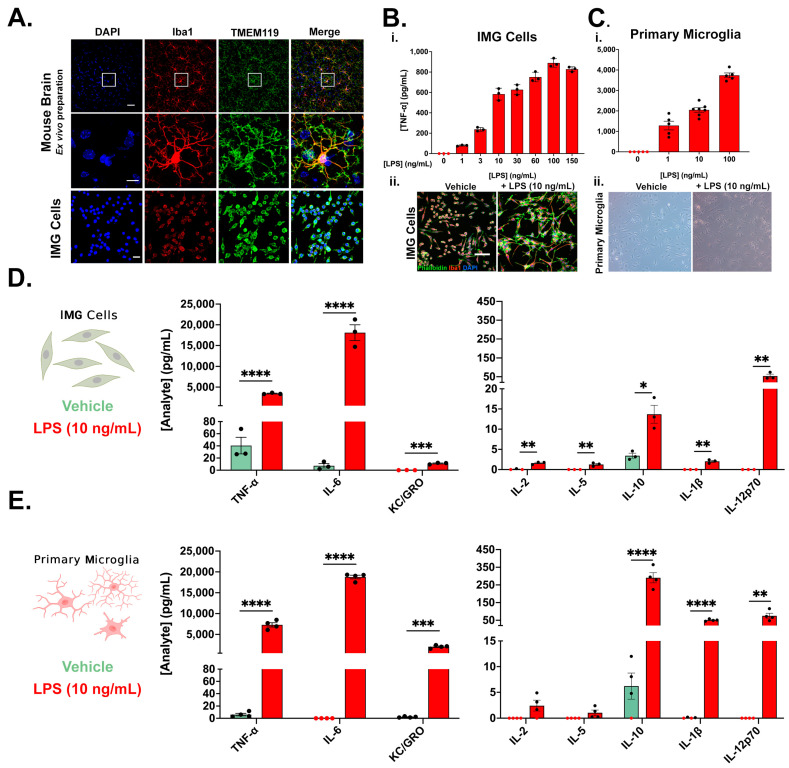
Immortalized mouse microglia (IMG cells) recapitulate features of primary microglia (PMg). (**A**) IMG cells express Iba1 and TMEM119, canonical markers of resident microglia of the mouse brain. The mouse brain images in the middle row are zoomed insets of the top row images, indicated by the white boxes in the image directly above. IMG cells (**B**(**i**)) (n = 3/group) and PMg (**C**(**i**)) (n = 5 or 7/group) secrete TNF-α in a dose-response manner to increasing doses of LPS 24 h after application. Additionally, representative morphological changes induced by a 10 ng/mL LPS challenge are similar between IMG cells ((**B**(**ii**))—via immunochemical staining) and PMg ((**C**(**ii**))—via light phase contrast microscopy). (**D**,**E**) Further analysis of the vehicle (0 ng/mL LPS)- and LPS (10 ng/mL)-treated cells from B revealed similar cytokine expression profiles in IMG cells (**D**) and PMg (**E**). LPS induces a significant increase in the production of TNF-α, IL-6, KC/GRO, IL-10, IL-1β, and IL-12p70 in both IMG cells (**D**) and PMg (**E**). Unpaired t-tests were used to compare vehicle vs. LPS-treated samples (* = *p* < 0.05, ** = *p* < 0.01, *** = *p* < 0.001, and **** = *p* < 0.0001); error bars represent mean ± SEM. Samples with analyte concentrations below assay detection limit are depicted in red and are assumed equivalent to 0 pg/mL to allow for statistical analysis. Brain microglia and IMG cells in (**A**) (scale bar = 20 μm); inset of single microglial cells in (**A**) (scale bar = 10 μm); IMG cells in (**B**(**ii**)) (scale bar = 50 μm). All ELISA experiments were normalized to MTS viability assays.

**Figure 2 cells-12-01367-f002:**
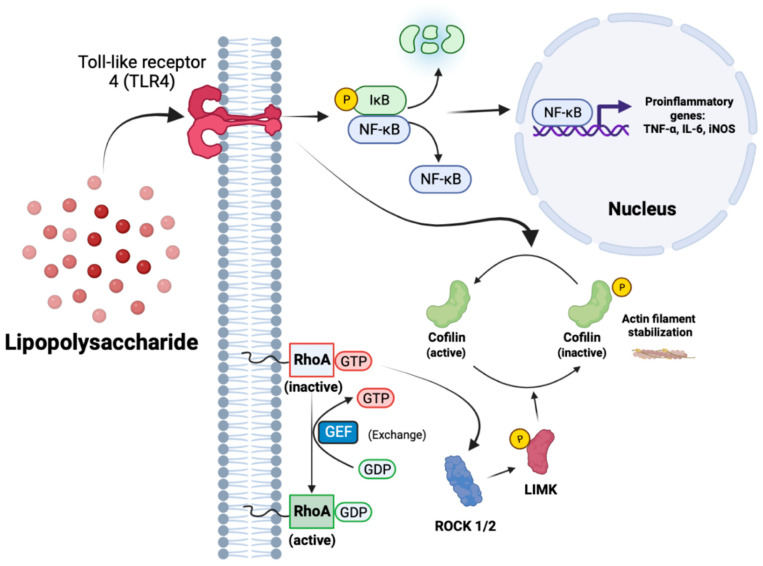
LPS (TLR4)-mediated inflammation and its interplay with RhoA activation and the downstream ROCK signaling pathways. LPS triggers the liberation of NF-κB from IκB, allowing for the NF-κB nuclear translocation and initiation of pro-inflammatory gene transcription. LPS also triggers the dephosphorylation of cofilin, which is the terminus of the RhoA/ROCK signaling pathway. Some of the effects of RhoA/ROCK signaling on inflammation remain elusive.

**Figure 3 cells-12-01367-f003:**
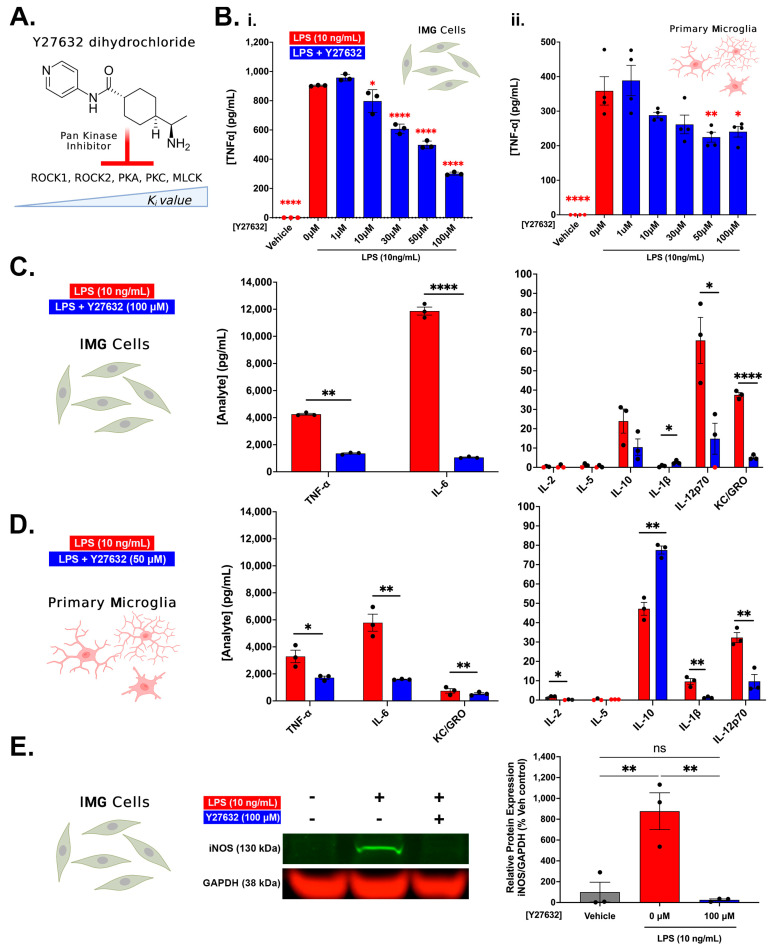
Pan kinase inhibition via Y27632 dihydrochloride reduces TLR4-initiated inflammatory response in IMG cells and PMg. (**A**) Chemical structure for Y27632 dihydrochloride. (**B**) Against an LPS challenge (10 ng/mL), TNF-α presence in media from IMG cells ((**B**(**i**)), n = 3/group) and PMg ((**B**(**ii**)), n = 4/group) decreases in a dose-response manner to increasing doses of Y27632. (**C**) Multiplex analyte analysis of Y27632 (100 μM) pre-treated IMG cells from (**B**) confirms significant reductions in TNF-α as well as the pro-inflammatory cytokine/chemokines IL-6, KC/GRO, and IL-12p70 when compared to LPS treatment alone. Y27632-pre-treated IMG cells show a minor, albeit significant, increase in the production of IL-1β. (**D**) Similar trends in Y27632 (50 μM)-pre-treated PMg from (**B**) (n = 3/group) were observed in the multi-analyte analysis. Significant TNF-α reductions were confirmed with equal or greater magnitude decreases in IL-6, IL-2, IL-1β, and IL-12p70 also observed. The anti-inflammatory cytokine IL-10 shows significant increases in the 50 μM pre-treated PMg. Whole-cell IMG cell lysates probed via Western Blot for iNOS (normalized to GAPDH protein) (**E**) show modulation of iNOS protein with LPS treatment and pretreatment with Y27632 (100 μM). Analyte samples with concentrations below the assay detection limit are shown as red data points and are assumed equivalent to 0 pg/mL to allow for the statistical analysis. In (**B**,**E**), one-way ANOVA with Dunnett’s comparison (vs. 0 μM Y27632) and Tukey’s comparison, respectively, were used. In (**C**,**D**), unpaired t-tests were used to compare LPS-treated vs. Y27632-pre-treated cells. Error bars represent mean ± SEM; (ns = not significant, * = *p* < 0.05, ** = *p* < 0.01, and **** = *p* < 0.0001). All ELISA experiments were normalized to MTS viability assays). Although the multiplex assay used in (**C**,**D**) also measured IL-4 and IFN-γ, no measurable analyte was observed and, hence, is not shown.

**Figure 4 cells-12-01367-f004:**
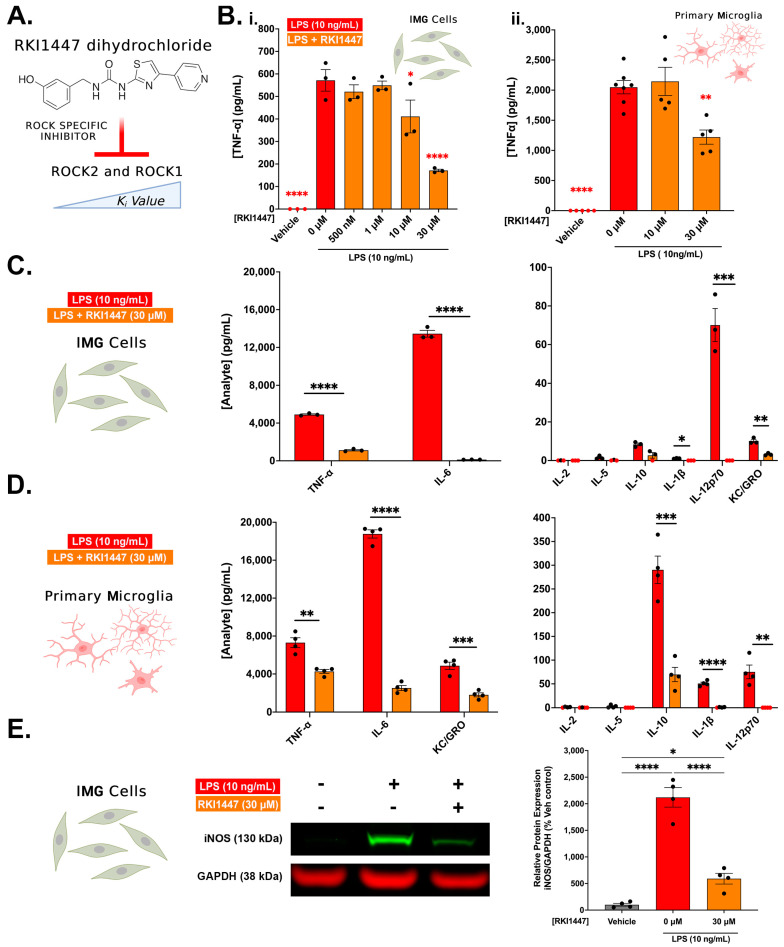
ROCK1 and ROCK2 inhibition via RKI1447 dihydrochloride is sufficient to block TLR4-initiated inflammatory response in IMG cells and PMg. (**A**) The chemical structure for the ROCK1 and ROCK2 specific inhibitor RKI1447 dihydrochloride is shown. (**B**) Against an LPS (10 ng/mL) challenge, TNF-α detected in in media from IMG cells ((**B**(**i**)), n = 3/group) and PMg ((**B**(**ii**)), n = 5 or 7/group) decreases in a dose-response manner to increasing pretreatment doses of RKI1447. (**C**,**D**) Multiplex analyte analysis of 30 μM pre-treated IMG cells from (**B**) confirms significant reductions in TNF-α as well as the pro-inflammatory cytokine/chemokines IL-6, KC/GRO, IL-12p70, IL-1β and the anti-inflammatory IL-10. (**D**) Similar trends in 30 μM pre-treated PMg from (**B**) (n = 3/group) were observed in the multiplex analyte analysis, confirming significant TNF-α reductions, along with significant decreases in IL-6, KC/GRO, IL-12p70, IL-1β, and the anti-inflammatory IL-10. Whole-cell IMG cell lysates probed via Western blotting for iNOS (normalized to GAPDH protein) (**E**) show significant increases in protein with LPS treatment and significant decreases in cells pre-treated with RKI1447. In (**B**,**E**), one-way ANOVA using Dunnett’s comparisons (vs. 0 μM RKI1447) and Tukey’s comparisons, respectively, were used. In (**C**,**D**), unpaired *t*-tests were used to compare LPS-treated vs. RKI1447-pre-treated cells. Analyte samples with concentrations below the assay detection limit are shown in red and are assumed equivalent to 0 pg/mL to allow for statistical analysis. Error bars represent mean ± SEM; (* = *p* < 0.05, ** = *p* < 0.01, *** = *p* < 0.001, and **** = *p* < 0.0001). All ELISA experiments were normalized to MTS viability assays. Although the multiplex assay used in (**C**,**D**) also measured IL-4 and IFN-γ, no measurable analyte was observed and hence is not shown.

**Figure 5 cells-12-01367-f005:**
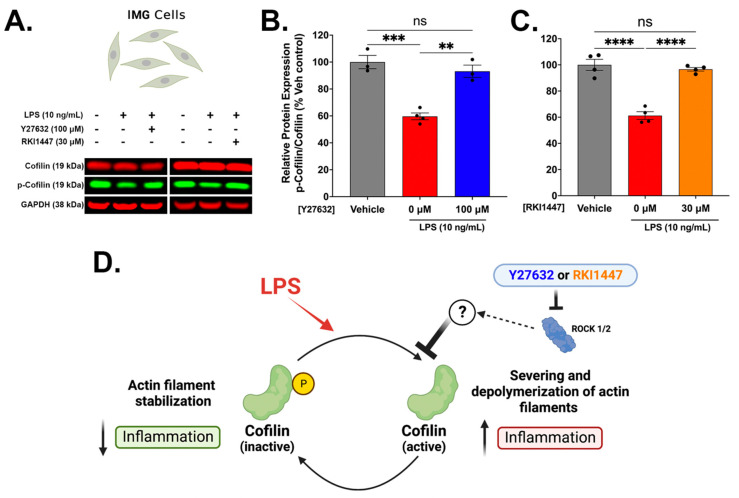
Y27632 or RKI1447 pretreatment in IMG cells blocks the dephosphorylation of cofilin. (**A**) Representative Western blots of whole-cell lysates of IMG cells 24 h after 10 ng/mL LPS challenge, with or without pretreatment of either Y27632 (100 μM) or RKI1447 (30 μM). Lysates were probed for *p*-cofilin (Ser-3), total cofilin, and GAPDH. (**B**,**C**) Quantification of the ratio of p-cofilin-to-cofilin in LPS vs. LPS + Y27632 (100 μM) (n = 3 or 4/group) (**B**) and LPS vs. RKI1447 (30 μM) (n = 4/group) (**C**). In both experiments, LPS significantly reduced the ratio of p-cofilin (inactive cofilin)-to-total cofilin. Y27632 and RKI1447 pretreatments restore the ratio to vehicle levels. (**D**) Schematic of cofilin activity induced by LPS challenge and inhibitory effects of Y27632 and RKI1447. One-way ANOVA with Tukey’s multiple comparisons was used for statistical analysis, with error bars representing mean ± SEM; ns = not significant, ** = *p* < 0.01, *** = *p* < 0.001, and **** = *p* < 0.0001.

**Figure 6 cells-12-01367-f006:**
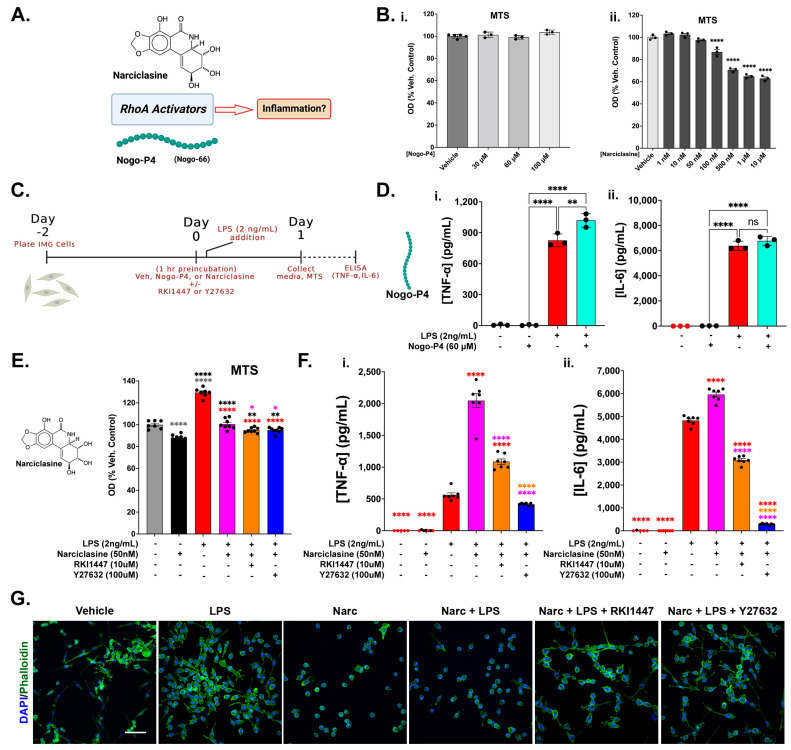
RhoA activation in IMG cells is not sufficient to induce an inflammatory response but exacerbates that response in the presence of a submaximal LPS challenge. (**A**) Narciclasine (Narc) and Nogo-P4 are RhoA activators. (**B**) MTS assays of IMG cells treated with Nogo-P4 (**i**) and Narc (**ii**) were used to assess the toxicity of various concentrations of each peptide/compound (n = 3/group). (**C**) Schematic of experimental setup for D-G. (**D**) TNF-α (**i**) and IL-6 (**ii**) production in IMG cells treated with Nogo-P4, LPS, and Nogo-P4 + LPS combination. (n = 3/group). (**E**) MTS viability assay for Narc (50 nM) ± LPS (2 ng/mL) and Narc (50 nM) + LPS (2 ng/mL) + Y27632 (100 μM) or RKI1447 (10 μM). (n = 7 or 8/group). MTS OD values from (**E**) are the same as the samples analyzed in (**F**) and were used to normalize data in (**i**) and (**ii**). (**F**) TNF-α and IL-6 ELISA data from the same treatments shown in (**E**). (**G**) Representative confocal images (40× magnification) of IMG cells stained for phalloidin (filamentous actin) and DAPI from all treatment groups from (**E**,**F**) (scale bar = 50 μm). One-way ANOVA was used for (**B**,**D**–**F**), with Dunnett’s and Tukey’s multiple comparison tests used, respectively, for the statistical analysis. Analyte samples with concentrations below the assay detection limit are shown in red and are assumed equivalent to 0 pg/mL to allow for statistical analysis. Error bars represent mean ± SEM. (ns = not significant, * = *p* < 0.05, ** = *p* < 0.01, and **** = *p* < 0.0001, with the color of the asterisks corresponding to the comparison group in (**E**,**F**)).

**Figure 7 cells-12-01367-f007:**
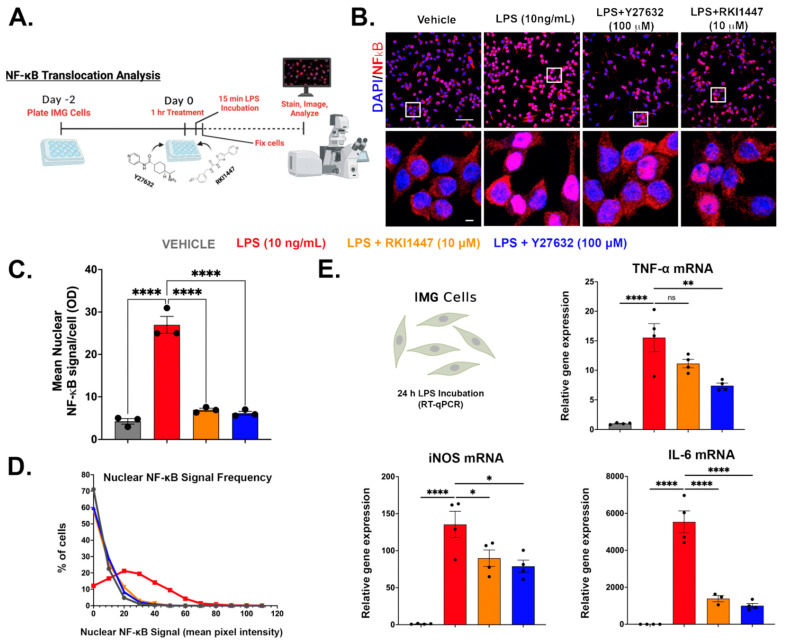
Y27632 and RKI1447 block NF-κB nuclear translocation and pro-inflammatory gene production in IMG cells. (**A**) Schematic of experimental timeline for (**B**–**D**). (**B**) Representative confocal images of NF-κB immunoreactivity (red) and DAPI (nuclei, blue) in IMG cells following LPS (10 ng/mL) challenge ± pretreatment of RKI1447 (10 μM) or Y27632 (100 μM). White boxes represent insets shown in the bottom panel. (**C**) Average nuclear NF-κB signal per cell for treatments shown in B (n = 3). (**D**) All cells analyzed in C were combined into a single histogram to represent the overall treatment effect. (**E**) Inflammatory gene production (TNF-α, IL-6, and iNOS) in IMG cells at 24 h post-LPS treatment (10 ng/mL) was reduced by RKI1447 (10 μM) and Y27632 (100 μM) pretreatment (n = 3 or 4/group). One-way ANOVAs were used in (**C**,**D**) using Tukey’s and Dunnett’s (vs. LPS treatment) multiple comparison tests, respectively. Error bars represent mean ± SEM (ns = not significant, * = *p* < 0.05, ** = *p* < 0.01, and **** = *p* < 0.0001).

**Figure 8 cells-12-01367-f008:**
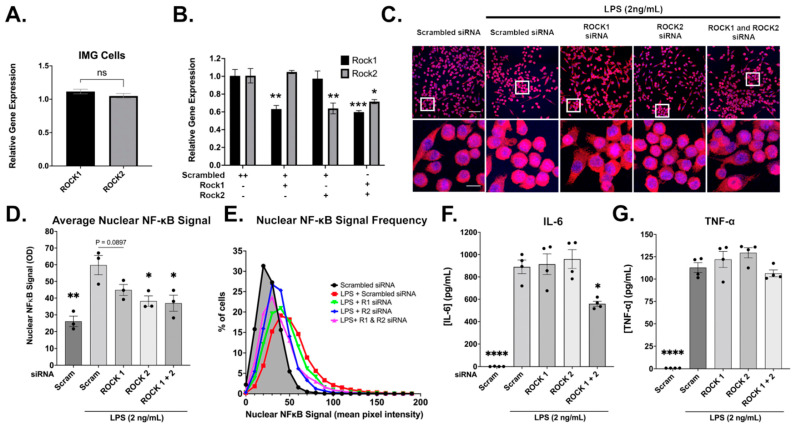
Both ROCK1 and ROCK2 inhibition may be necessary to block LPS-initiated inflammatory response. (**A**) ROCK1 and ROCK2 expression in IMG cells (normalized to β-actin expression). (**B**) ROCK1 and ROCK2 siRNAs are highly specific, significantly reducing their target genes (n = 3/group). (**C**) Representative confocal images of NF-κB immunoreactivity (red) and DAPI (blue) in IMG cells following pretreatment with 5 pmol (10 pmol total in each treatment) of their respective siRNA (ROCK1 + Scrambled, ROCK2 + Scrambled, ROCK 1 + ROCK 2, or Scrambled + Scrambled) ± LPS (2 ng/mL). The white boxes in the top panel are shown full size in the bottom panel. (**D**) Average nuclear NF-κB signal per cell for treatments shown in B (n = 3). (**E**) All cells analyzed in C were combined into a single histogram to represent the overall treatment effect. Media from treatment groups were analyzed for IL-6 (**F**) and TNF-α (**G**). ROCK1 and ROCK2 siRNA treatment significantly reduced IL-6 protein production from the IMG cells. For the comparisons in B, two-way ANOVAs with Tukey’s multiple comparison tests were used, and in (**D**,**F**,**G**), one-way ANOVAs with Dunnett’s (vs. Scrambled + LPS treatment) multiple comparison tests were used for statistical analyses. Error bars represent mean ± SEM (ns = not significant, * = *p* < 0.05, ** = *p* < 0.01, *** = *p* < 0.001, and **** = *p* < 0.0001).

**Figure 9 cells-12-01367-f009:**
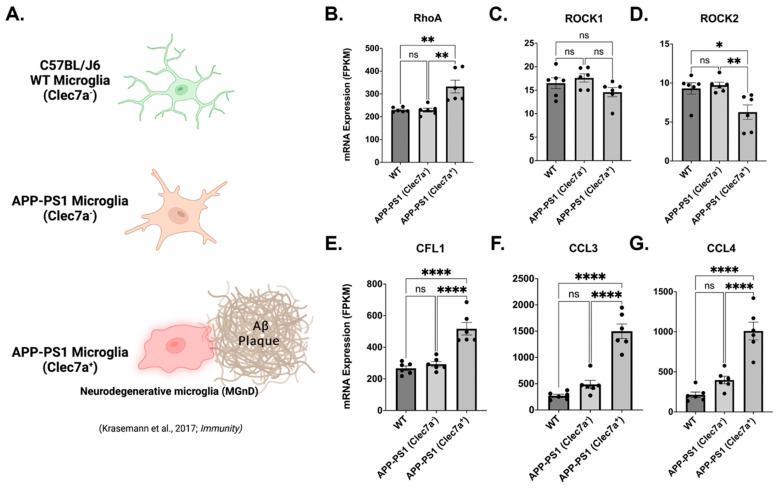
Neurodegenerative microglia (MGnD) from aged (9 mo) APP/PS1 Alzheimer’s disease mice upregulate key inflammatory and RhoA signaling genes. Data from [41]. (**A**) WT microglia (from C57BL/J6 mice), Clec7a^−^ microglia (from APP/PS1 mice), and Clec7a^+^ neurodegenerative microglia (MGnD) (from APP/PS1 mice) differ in morphology. MGnD is associated with neuritic amyloid-β plaques, while Clec7a^−^ microglia from the same mice do not. (**B**–**E**) Expression of fundamental genes involved in the RhoA signaling pathway (RHOA, ROCK1, ROCK2, CFL1) and (**F**,**G**) pro-inflammatory regulators (CCL3 and CCL4) are shown from the aforementioned microglia subtypes (n = 6/group). One-way ANOVAs were used with Tukey’s multiple comparison tests. Error bars represent mean ± SEM (ns = not significant, * = *p* < 0.05, ** = *p* < 0.01, and **** = *p* < 0.0001).

**Table 1 cells-12-01367-t001:** Primer sequences for primers used in the evaluation of mRNA transcript levels.

Probe Name		5′-3′ Sequence	Source
ROCK1	Fw	GCTCATCTCTGTGTGACTCT	NM_009071.2
	Rv	TACGGAAAGCAAGTCAGACC	
ROCK2	Fw	GGTCAATCAGCTCCAGAAAC	NM_009072.2
	Rv	GTTTGGAACTTTCTGCCTGG	
TNF-α	Fw	GGCAGGTCTACTTTGGAGTCATTG	[49]
	Rv	ACATTCGAGGCTCCAGTGAATTCGG	
iNOS	Fw	TACTCCATCAGCTCCTCCCA	NM_010927.4
	Rv	GTTCCTGATCCAAGTGCTGC	
IL-6	Fw	TTCTTGGGACTGATGTTGTTGAC	[50]
	Rv	AATTAAGCCTCCGACTTGTGAAG	
β-actin	Fw	TGAGAGGGAAATCGTGCGTGAC	NM_007393.4
	Rv	CCGCTCGTTGCCAATAGTGATG	

## Data Availability

The data that support the findings of this study are available from the corresponding author upon reasonable request. Supporting data can also be found in Supplementary figures.

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
