# Peer review of "The RhoA-ROCK1/ROCK2 Pathway Exacerbates Inflammatory Signaling in Immortalized and Primary Microglia"

_cells, 2023, doi:10.3390/cells12101367_

Round 1

Reviewer 1 Report

This study reports the role of ROCK1 and ROCK2 in the neuroinflammation. They have showed that the inhibitors of ROCK1 and ROCK2 inhibition regulates the inflammation via NFkB translocation and cofilin activation. Authors utilized two different cell model to prove the hypothesis in this study. However this study has some errors and many inconsistencies and its need to be addressed before considering for the publication. Please address the following concerns,

Authors mentioned both cell line models in the title, but they used majorly IMG and PMG used in very few sections. Also, why the authors decided to use IMG and PMG. PMG alone should be sufficient and that’s a better model than IMG. Overall, the usage of model in the all the sections could be improved.

I suggest the authors to keep the color of the bar graph consistent. Ex. Blue for LPS group then use blue throughout the manuscript.

Why authors performed western blots only in IMG and not in PMG since they used other experiments in both models?

Fig 3C why IL-10 expression so high in IMG while low expression in PMG with LPS treatment? Is LPS induces the anti-inflammatory genes in IMG? Is any other paper reported this?

Fig 3C & D include the vehicle group and do the stats again.

Fig 4C & D include the vehicle group and do the stats again.

Fig 3E&4E why there is huge difference in the INOS expression in LPS even though its same treatment condition?

Fig 5A please mention the phosphorylation site for the Cofilin. Please show the expression levels of Cofilin and p-Cofilin in the PMG. Please be consistent with the experiment plan.

Fig 8B is not clear, is authors used Scramble and Rock siRNA together? And why?

Why the LPS concentration is not consistent with the experiments? Some experiments using 10ng/ml and some places 2ng/ml were used. The concentration should be remained same for the all the experiments.

Authors showed degenerative microglia express increased inflammatory genes which is associated with amyloid beta plaques and isolated from AppPs1 transgenic mice. Please show the treatment with synthetic amyloid beta and their effect of the inflammatory mediators expression. Also why authors showed only the chemokine levels and not the cytokine levels in the MGnD?

Is the ROCK inhibitors had any effect on the inflammatory mediators expression in the MGnD? Please explain.  

Reviewer 2 Report

Glotfelty et al. compare the interactions between RhoA-ROCK1/2 and LPS-induced NF-kB signaling pathways in IMG cell line and primary mouse microglia. Activation of the RhoA-ROCK1/2 pathway exacerbates inflammatory response of IMG and primary microglia challenged with LPS. Inhibition of the ROCK1/2 activity results in reduced secretion of pro-inflammatory cytokines and restores the phospho-cofilin levels. The manuscript is clearly written, results are presented in a logical and clear way. There are several minor points that need addressing:

1.    Shouldn’t the volume of media used for culturing cells be expressed in microliters (µl) pre well instead of milliliters (mL) per well?

2.    The results from the multiplex cytokine measurements where the analytes were below detection level should be shown in supplementary material.

3.    The sentence in lines 522-523 needs editing.

4.    Figures 3C-D and 4C-D: the graphs showing concentrations of various cytokines should be grouped according to their concentration to avoid using split y-axes. The observed differences in cytokine concentrations look confusing when graphs with split y-axes are used and often look smaller than the actual differences leading this reviewer and readers to initially question the biological significance of the data.

5.    The sentence in lines 650-652 needs editing.

6.    Figures 7D and 8E: the overlapping distributions of cell percentages with nuclear NF-kB signal are shown poorly in the current graphs. Use of transparent colors under curves or no colors would make the data easier to interpret.

Round 2

Reviewer 1 Report

I appreciate the authors for the detailed responses. I am pretty much satisfied with all the responses. I just have a minor comment.

The authors mentioned in the response that TNF alpha levels are below the detection level. In that case the analytes were below detection level and should be shown in supplementary material.
